# MeZO-A³dam: Memory-efficient Zeroth-order Adam with Adaptivity Adjustments for Fine-tuning LLMs

## Abstract

Recently, fine-tuning of language models (LMs) via zeroth-order (ZO) optimization has gained significant traction due to their ability of memory-efficient deployment, significantly reducing memory cost over first-order methods. However, the existing studies on ZO optimization for LM fine-tuning often exhibit slow convergence and reliance on the hand-crafted prompts. In this paper, we first investigate the importance of the adaptive gradient-based ZO optimization method in mitigating these limitations. Toward this, we revisit memory-efficient zeroth-order Adam (MeZO-Adam) and make important findings that merely considering adaptivity can enable faster convergence while improving the generalization ability compared to previous studies. Interestingly, we further observe that decreasing the level of adaptivity might be recommended in ZO optimization potentially due to the high variance of ZO gradient estimate, hypothesized as *weak adaptivity hypothesis*. Based upon our hypothesis, we propose **MeZO-A³dam**, MeZO-Adam with Adaptivity Adjustments according to the parameter dimension. We provide the dimension-free theoretical guarantee on both the convergence and the generalization of MeZO-A³dam, providing strong evidence for our hypothesis. Extensive experiments show that MeZO-A³dam can achieve faster convergence and better generalization over several baselines across LMs of various sizes on diverse datasets. By adaptivity adjustments, MeZO-A³dam outperforms MeZO, MeZO-SVRG, and MeZO-Adam, with up to an average of $36.6\%$, $16.9\%$, $6.8\%$ improvements in performance and up to an average of $\times 12.6$ and $\times 1.8$ faster convergence, respectively. Furthermore, by leveraging an off-the-shelf low-bit optimizer, MeZO-A³dam achieves an average of $40.3\%$ and $43.6\%$ memory reduction from MeZO-SVRG and MeZO-Adam.

## 1 Introduction

Since the rise of large language models (LLMs), research has focused on leveraging their capabilities (Radford et al., 2019; Zhang et al., 2022; Touvron et al., 2023). Building on their success, fine-tuning pre-trained LLMs becomes a key strategy for adapting them to downstream tasks. However, it requires significant memory, making it impractical for practitioners. This challenge has led to alternatives like in-context learning (ICL, Min et al. (2022)) and parameter-efficient fine-tuning (PEFT, Li & Liang (2021); Lester et al. (2021); Hu et al. (2022)). However, ICL often yields suboptimal performance (Malladi et al., 2023) and requires time-consuming prompt engineering, introducing memory overhead. While PEFT is more efficient than full fine-tuning (Full FT), it still requires more memory than inference alone due to activation memory (Azizi et al., 2024) and one may increase the number of trainable parameters to achieve satisfactory performance, potentially driving up resources.

As a consequence, a separate line of research, the zeroth-order (ZO) optimization recently have been explored to fine-tune LLMs (Malladi et al., 2023; Guo et al., 2024; Liu et al., 2024b; Zhang et al., 2024). ZO optimization requires neither the memory nor the computation costs associated with gradient calculations. Additionally, unlike ICL, it does not require longer context length. The pioneering study, MeZO (Malladi et al., 2023) demonstrated that zeroth-order stochastic gradient descent (SGD) (Robbins & Monro, 1951) can effectively fine-tune language models with only inference-time memory and computation budget.

Table 1: Comparisons among SoTA memory-efficient ZO methods with the parameter dimension $d$. The symbols ✓/✗ indicate whether each column is considered or not in the analysis of each method. In summary, our work is the first attempt to analyze the convergence and generalization considering both the momentum and the preconditioner in terms of theory. Also, the dimension-free guarantees are significant contributions in theory, which is generally not available in previous ZO literature.

| Algorithm | Momentum | Preconditioner | Convergence Guarantee | Generalization Error Bound |
|---|---|---|---|---|
| MeZO | ✗ | ✗ | ✓(Dimension-free) | ✗ |
| MeZO-SVRG | ✗ | ✗ | ✓(Depends on $d$) | ✗ |
| MeZO-A$^3$dam (**Ours**) | ✓ | ✓(Adam) | ✓(Dimension-free) | ✓(Dimension-free) |

However, while MeZO reduces memory consumption, it suffers from slower convergence and longer training time. Also, MeZO requires hand-crafted, task-specific prompts to achieve acceptable performance, which limits its broader applicability. While prompts are common in LLM fine-tuning (Wei et al., 2022; Sanh et al., 2022; Ouyang et al., 2022; Chung et al., 2024), their complexity diminishes the practicality of ZO optimization. To speed up convergence, sparse gradient methods (Liu et al., 2024b; Guo et al., 2024) have been proposed, selectively updating a small subset of parameters, though prompts are still required. More recently, MeZO-SVRG (Gautam et al., 2024) integrates the stochastic variance-reduced gradient (SVRG) estimator for faster convergence without prompts, however, it incurs high computational costs due to full-batch gradient calculations. In addition, the variance-reduced method may not be effective for deep learning tasks (Defazio & Bottou, 2019). Furthermore, their accuracy gap compared to full FT still remains to be often significant.

All the aforementioned studies have focused on vanilla SGD while adaptive gradient methods such as Adam (Kingma & Ba, 2015; Loshchilov & Hutter, 2019) or Lion (Chen et al., 2023) have become standard in traditional fine-tuning of LLMs. In MeZO, only a brief introduction to MeZO-Adam and simple experimental results in prompt-dependent settings have been presented, showing little difference from MeZO. However, the full potential of adaptive gradients in more generalized ZO fine-tuning scenarios is yet to be fully unveiled.

Toward mitigating the above limitations of MeZO, in this paper, we revisit the MeZO-Adam in LLM fine-tuning. We first empirically discover that the adaptive gradients are advantageous in a prompt-free fine-tuning scenario. More importantly, we observe that controlling the amount of adaptivity of MeZO-Adam indeed matters, upon which we introduce ***weak adaptivity hypothesis*** and propose ***MeZO-A$^3$dam*** that allows to adjust the adaptivity according to the parameter dimension.

Our main contributions are summarized below:

- We thoroughly investigate on the adaptive gradient ZO optimization for fine-tuning and first reveal its importance in a prompt-free fine-tuning scenario. Specifically, we make an important observation that *decreasing the adaptivity level can potentially be beneficial in ZO optimization*. This may be attributed to the high variance of ZO gradient estimate. We then articulate *weak adaptivity hypothesis* and present MeZO-A$^3$dam. This allows the degree of adaptivity according to the parameter dimension enabling faster convergence compared to non-adaptive alternatives.

- We theoretically analyze the convergence and the generalization of MeZO-A$^3$dam. While the convergence complexity and the generalization error bound heavily depend on the parameter dimension $d$ in most ZO optimization analysis, we make an important theoretical observation that ***decreasing the adaptivity level can allow us to obtain a dimension-free guarantee***. This provides strong evidence for the weak adaptivity hypothesis. Additionally, we provide theoretical insights into *how much the degree of adaptivity should be adjusted*. Table 1 summarizes the comparisons in theory among state-of-the-art memory-efficient zeroth-order methods.

- Our extensive experiments demonstrate that MeZO-A$^3$dam consistently surpasses several existing ZO baselines across various sizes of language models on different benchmarks, in all three aspects: generalization, convergence, and memory consumption. More precisely, MeZO-A$^3$dam outperforms (1) {MeZO, MeZO-SVRG, MeZO-Adam} with up to an average of {36.6%, 16.9%, 6.8%} *better generalization*, (2) {MeZO-SVRG, MeZO-Adam} with an average of {×12.6, ×1.8} *faster convergence speed*, and (3) {MeZO-SVRG, MeZO-Adam} with an average of {40.3%, 43.6%} *lower memory consumption*, respectively.

**Algorithm 1** MeZO-A[3]dam: **M**emory-**e**fficient **Z**eroth-**O**rder **Adam** with Adaptivity Adjustment
___
1: **Input:** Stepsize $\alpha_t$, momentum parameter $(\beta_1, \beta_2) = (0.9, 0.999)$, batch size $B$, base adaptivity parameter $\delta_0 = 10^{-8}$, and scaling function $h : \mathbb{R} \to \mathbb{R}$ (ex. $h(d) = d$).
2: **Initialize:** $\boldsymbol{\theta}_0 \in \mathbb{R}^d$, $\boldsymbol{m}_0 = 0$, and $\boldsymbol{V}_0 = 0$.
3: **for** $t = 1, 2, \ldots, T$ **do**
4:      Draw a minibatch sample $\mathcal{S} = \{\boldsymbol{x}_1, \cdots, \boldsymbol{x}_B\}$.
5:      $\boldsymbol{g}_t \leftarrow \frac{1}{B}\sum_{i=1}^{B} \widehat{\nabla}\mathcal{L}(\boldsymbol{\theta}_{t-1}; \boldsymbol{z}_i).$            ▷ Minibatch ZO gradient (Definition 2.1)
6:      $\boldsymbol{m}_t \leftarrow \beta_1 \boldsymbol{m}_{t-1} + (1 - \beta_1)\boldsymbol{g}_t.$            ▷ Momentum construction
7:      $\boldsymbol{v}_t \leftarrow \beta_2 \boldsymbol{v}_{t-1} + (1 - \beta_2)\boldsymbol{g}_t^{\odot 2}.$          ▷ Preconditioner construction
8:      $\widehat{\boldsymbol{m}}_t, \widehat{\boldsymbol{v}}_t \leftarrow \frac{\boldsymbol{m}_t}{1 - \beta_1^t}, \frac{\boldsymbol{v}_t}{1 - \beta_2^t}$           ▷ Bias corrections
9:      $\delta \leftarrow \delta_0 h(d)$                          ▷ Adaptivity Adjustment
10:     $\boldsymbol{\theta}_t \leftarrow \boldsymbol{\theta}_{t-1} - \alpha_t \frac{\widehat{\boldsymbol{m}}_t}{\sqrt{\widehat{\boldsymbol{v}}_t} + \delta}$            ▷ Descent step
11: **end for**
12: **Output:** $\theta_T$
___

## 2 PRELIMINARY: MEMORY-EFFICIENT ZEROTH-ORDER OPTIMIZATION

In this section, we introduce the memory-efficient zeroth-order optimization (MeZO), a pioneering study in this line of work. We describe three components of MeZO, namely, (i) simultaneous perturbation stochastic approximation (SPSA), (ii) zeroth-order SGD (ZO-SGD), and (iii) memory-efficient implementation.

**Definition 2.1** (Simultaneous Perturbation Stochastic Approximation (SPSA, Spall (1992))). *For a model parameter $\boldsymbol{\theta} \in \mathbb{R}^d$ and a loss $\mathcal{L}$, the SPSA on minibatch $\mathcal{B}$ estimates the gradient as*

$$\widehat{\nabla}\mathcal{L}(\boldsymbol{\theta}; \mathcal{B}) = \frac{\mathcal{L}(\boldsymbol{\theta} + \mu\boldsymbol{u}; \mathcal{B}) - \mathcal{L}(\boldsymbol{\theta} - \mu\boldsymbol{u}; \mathcal{B})}{2\mu}\boldsymbol{u}$$

*where the random vector $\boldsymbol{u}$ is sampled from standard $d$-dimensional Gaussian distribution $\boldsymbol{u} \sim \mathcal{N}(\boldsymbol{0}, \boldsymbol{I}_d)$ and $\mu$ being a smoothing parameter.*

Note that, under the limit case of $\mu \to 0$, the SPSA gradient estimate becomes an unbiased estimator of the first-order true gradient $\nabla\mathcal{L}(\boldsymbol{\theta})$.

**Zeroth-Order SGD (ZO-SGD).** The zeroth-order SGD updates the model parameter $\theta_t$ via ZO gradient estimate as $\boldsymbol{\theta}_{t+1} = \boldsymbol{\theta}_t - \alpha_t \widehat{\nabla}\mathcal{L}(\boldsymbol{\theta}_t; \mathcal{B}_t)$ where $\alpha_t$ is the stepsize at time $t$ and $\mathcal{B}_t$ is the minibatch at time $t$. Note that ZO-SGD must save the variable $u$ in the memory since the same random variable $u$ should be used in computing both $\mathcal{L}(\boldsymbol{\theta} + \mu\boldsymbol{u})$ and $\mathcal{L}(\boldsymbol{\theta} - \mu\boldsymbol{u})$.

**Memory-efficient Implementation.** In extremely high-dimensional problems, storing the variable $\boldsymbol{u}$ and the gradient $\widehat{\nabla}\mathcal{L}(\boldsymbol{\theta})$ requires additional memory equivalent to the model parameters, which can impose a burden in terms of memory. To bypass caching the variable $\boldsymbol{u}$, Malladi et al. (2023) propose an in-place implementation by storing a single random seed and reproducing the variable $\boldsymbol{u}$ when required. We provide the pseudocode for the detailed implementations including MeZO in Appendix.

## 3 CONTROLLING ADAPTIVITY MATTERS IN ZEROTH-ORDER OPTIMIZATION

In this section, we discover the importance of adaptive gradients in zeroth-order fine-tuning. Furthermore, we highlight the significance of handling the adaptivity level, through which we articulate *weak adaptivity hypothesis* and propose our optimization algorithm, MeZO-A[3]dam.

### 3.1 MEMORY-EFFICIENT ZEROTH-ORDER ADAM (MEZO-ADAM)

Although MeZO has become a breakthrough approach in LLM fine-tuning due to its memory efficiency, it suffers from notoriously slow convergence, which takes excessive overall fine-tuning time. To address this issue, MeZO-SVRG appears to mitigate convergence speed through a variance-reduced method, but the average per-step time still remains considerably slow in practice due to the

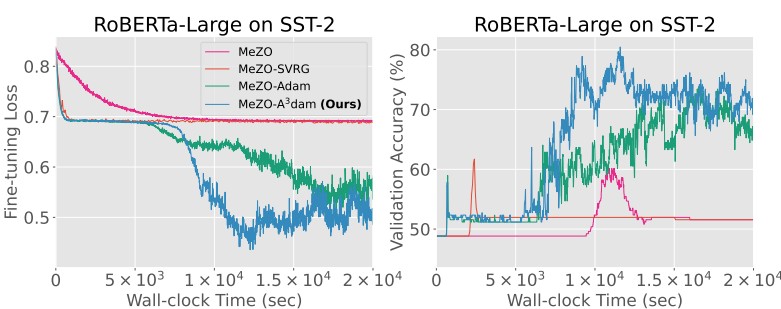

Figure 1: A proof-of-concept: fine-tuning RoBERTa-Large on SST-2 with ZO methods.

full-batch gradient computation. Moreover, the variance-reduced method may not be effective for deep learning tasks according to the previous study (Defazio & Bottou, 2019).

In light of these factors, also as adaptive gradient methods have become standard in LLM fine-tuning, we begin by questioning *the value of employing adaptive gradients in zeroth-order optimization*. Toward this, in this paper, we revisit the memory-efficient zeroth-order version of Adam (MeZO-Adam). We provide the detailed algorithm in Alg. 1, in which $\boldsymbol{m}_t$ and $\boldsymbol{v}_t$ are constructed in the same manner as the first-order Adam (line $5 \sim 8$) except the gradient estimation (line 5). The scaling function $h$ (line 9) for MeZO-Adam can be simplified to $h(d) = 1$.

Note that, in Alg. 1, the construction of $\boldsymbol{m}_t$ and $\boldsymbol{v}_t$ could be implemented in a memory-efficient manner, however, we empirically observe that it is not computationally efficient. Therefore, in practice, we merely construct the ZO gradient at each iteration and save only $\boldsymbol{m}_t$ and $\boldsymbol{v}_t$ in the optimizer states. Note also that the small constant $\delta$, added to $\sqrt{\boldsymbol{v}_t}$ to prevent the denominator from being zero, is referred to as an *adaptivity parameter*. If $\delta$ is large, the influence of the preconditioner $\sqrt{\boldsymbol{v}_t}$ diminishes, allowing $\delta$ to control the level of adaptivity. There have been studies on the role of $\delta$ (Zaheer et al., 2018; Nado et al., 2021), however, it has never been explored in ZO optimization. Throughout this paper, we refer to $\delta$ as the adaptivity parameter.

### 3.2 MeZO-A³dam: MeZO-Adam with Adaptivity Adjustments under Weak Adaptivity Hypothesis

In this study, the adaptivity parameter $\delta$ plays a crucial role, which will be discussed in this section. Note that it is known that stochastic ZO gradient exhibits high variance which comes from both (i) high dimensionality of the parameter and (ii) minibatch sampling. Notably, sampling noise is also inherent in first-order stochastic gradients; thus, the primary factor for the high variance of stochastic ZO gradient results from high dimensionality. Based on this observation, in zeroth-order optimization, we hypothesize that the base adaptivity parameter $\delta_0 = 10^{-8}$ of Adam might encourage excessive adaptivity where the ZO gradient estimate already has a high variance, which can eventually hinder the optimization process. Under this intuition, we articulate the *weak adaptivity hypothesis*.

**Weak Adaptivity Hypothesis.** *The adaptivity parameter of zeroth-order Adam should be scaled according to the parameter dimension relative to the base adaptivity parameter of first-order Adam.*

In other words, this hypothesis means that the *small amount of adaptivity is enough* in ZO optimization. The hypothesis directly motivates our proposed algorithm, **MeZO-A³dam**, which is **MeZO-Adam with Adaptivity Adjustments**. In details, the line 9 with a non-trivial scaling function $h$ in Alg. 1 characterizes the key feature of MeZO-A³dam.

As a proof-of-concept, we explore the optimization of training RoBERTa-Large (Liu, 2019) on SST-2, focusing on two main points: (i) whether using adaptive gradients in zeroth-order optimization can outperform existing ZO methods such as MeZO or MeZO-SVRG; (ii) whether scaling up $\delta$ in MeZO-Adam actually provides any benefits in terms of optimization. Figure 1 illustrates our proof-of-concept results on the learning curves of both fine-tuning loss and validation accuracy. As MeZO-Adam itself surpasses ZO baselines in terms of both convergence and generalization, the advantages of using adaptive gradients can be clearly observed. More surprisingly, MeZO-A³dam which adjusts the adaptivity with $\delta = 10^{-2}$ shows faster convergence and achieves lower fine-tuning loss as well as better generalization than MeZO-Adam, which partially supports our hypothesis.

In the following section, we aim to address our hypothesis via in-depth study from a theoretical perspective. Additionally, we intend to provide theoretical insights on *"how to choose the scaling function h in practice?"*. In other words, *"how much the adaptivity parameter δ should be scaled?"*

# 4 THEORY

In this section, our primary goal is to provide the theoretical evidence for weak adaptivity hypothesis in terms of both convergence and generalization in ZO optimization. We are interested in the optimization problem: $\min_{\boldsymbol{\theta} \in \mathbb{R}^d} \mathcal{L}(\boldsymbol{\theta}) \coloneqq \frac{1}{n} \sum_{i=1}^{n} \mathcal{L}(\boldsymbol{\theta}; \boldsymbol{z}_i)$ for dataset $S = \{\boldsymbol{z}_1, \cdots, \boldsymbol{z}_n\}$ where $\mathcal{L}(\boldsymbol{\theta}; \boldsymbol{z}_i) =: \mathcal{L}_i(\boldsymbol{\theta})$ denotes the loss evaluated on the single datapoint $\boldsymbol{z}_i$. For notations, we simply use $\|\cdot\|$ for $\ell_2$-norm and $\|A\|_2$ represents the matrix 2-norm, i.e., the maximum eigenvalue of a matrix $A$.

## 4.1 CONVERGENCE

We analyze the convergence of MeZO-Adam in non-convex optimization. For this purpose, we begin with the following standard conditions in this line of research.

**(C-1)** The loss function $\mathcal{L}$ is $L$-smooth, i.e., $\|\nabla \mathcal{L}(\boldsymbol{\theta}) - \nabla \mathcal{L}(\boldsymbol{\theta}')\| \leq L \|\boldsymbol{\theta} - \boldsymbol{\theta}'\|$ for all $\boldsymbol{\theta}, \boldsymbol{\theta}' \in \mathbb{R}^d$.

**(C-2)** The first-order stochastic gradient is unbiased and has a bounded variance. Further, we assume that the true gradient is bounded, i.e., $\mathbb{E}\left[\nabla \mathcal{L}(\boldsymbol{\theta}; \boldsymbol{z})\right] = \nabla \mathcal{L}(\boldsymbol{\theta})$, $\mathbb{E}\left[\|\nabla \mathcal{L}(\boldsymbol{\theta}) - \nabla \mathcal{L}(\boldsymbol{\theta}; \boldsymbol{z})\|^2\right] \leq \sigma^2$, and $\|\nabla \mathcal{L}(\boldsymbol{\theta})\| \leq G$ for all datapoint $\boldsymbol{z}$.

**(C-3)** There exists $\gamma \geq 0$ and $\zeta \geq 0$ such that

$$\gamma \leq \min_{i \in [d], t \in [T]} \boldsymbol{v}_{t,i}, \quad \zeta \geq \max_{i \in [d], t \in [T]} \boldsymbol{v}_{t,i},$$

where $\boldsymbol{v}_{t,i}$ denotes the $i$-th coordinate of $\boldsymbol{v}_t$. We define a condition number $\kappa_\delta$ which is the key quantity for our analysis: $\kappa_\delta = \frac{\sqrt{\zeta}+\delta}{\sqrt{\gamma}+\delta}$.

The condition (C-1) is standard in non-convex optimization analysis. In condition (C-2), the unbiasedness of stochastic gradients and bounded variance are fundamental in stochastic optimization literature (Ghadimi & Lan, 2013). The bounded gradient condition in (C-2) is frequently used in convergence analysis in the context of adaptive gradient methods (Kingma & Ba, 2015). The last condition (C-3) is also assumed in the previous study on the analysis of zeroth-order adaptive gradient methods (Chen et al., 2019; Nguyen et al., 2022).

Along with standard conditions, we introduce our key assumption to achieve dimension-free convergence. Toward this, we review the *local effective rank* condition proposed in Malladi et al. (2023).

**Assumption 4.1** (Local Effective Rank Malladi et al. (2023))**.** *Let* $G(\theta_t) \coloneqq \max_{i \in [n]} \|\nabla \mathcal{L}(\boldsymbol{\theta}_t; \boldsymbol{z}_i)\|$ *where* $\boldsymbol{z}_i$ *is a data sample from training dataset* $S = \{\boldsymbol{z}_1, \cdots, \boldsymbol{z}_n\}$. *Then, there exists a matrix* $H(\theta_t) \preceq L \cdot \boldsymbol{I}_d$ *for the $L$-smooth loss such that:*

*1. For all $\boldsymbol{\theta}$ satisfying $\|\boldsymbol{\theta} - \boldsymbol{\theta}_t\| \leq \alpha d G(\boldsymbol{\theta}_t)$, we have $\nabla^2 \mathcal{L}(\boldsymbol{\theta}) \preceq H(\boldsymbol{\theta}_t)$.*

*2. The effective rank of $H(\boldsymbol{\theta}_t)$, i.e., $\frac{\text{Tr}(H(\boldsymbol{\theta}_t))}{\|H(\boldsymbol{\theta}_t)\|_2}$ is at most $r$ where $r \ll d$.*

The authors assume that $d$-dimensional Gaussian random variable is sampled from the radius of $\sqrt{d}$ for simplicity, which we also assume for our theory, however, the analysis could be easily extended to a general Gaussian random variable case using the probabilistic approaches (i.e. given failure probability $\eta \in (0, 1)$, the statement holds with probability at least $1 - \eta$). The Assumption 4.1 enables the convergence rate to be irrelevant of the problem dimension $d$ by dramatically reducing the number of directions that the model parameter could move along around the current parameter $\boldsymbol{\theta}_t$. Note that, however, Malladi et al. (2023) only consider the limit case of the smoothing parameter $\mu$, i.e., $\mu \to 0$, while the zeroth-order algorithm we actually use in practice employ strictly positive $\mu > 0$, thus the analysis in MeZO is not available in real cases. In addition, the analysis in Malladi et al. (2023) do not allow for the adaptive gradients, such as ADAGRAD or ADAM. To bridge the gap in the theory, we propose the *revised version of local effective rank condition* that allows for both strictly positive smoothing parameter $\mu > 0$ and the adaptive gradients.

**Assumption 4.2** (Revised Local Effective Rank). *Let $G(\boldsymbol{\theta}_t) \coloneqq \max_{i \in [n]} \|\nabla \mathcal{L}(\boldsymbol{\theta}_t; \boldsymbol{z}_i)\|$ where $\boldsymbol{z}_i$ is a data sample from training dataset $S = \{\boldsymbol{z}_1, \cdots, \boldsymbol{z}_n\}$. Then, there exists a matrix $H(\boldsymbol{\theta}_t) \preceq L \cdot \boldsymbol{I}_d$ for the $L$-smooth loss such that:*

*1. For all $\boldsymbol{\theta}$ satisfying $\|\boldsymbol{\theta} - \boldsymbol{\theta}_t\| \leq \frac{\alpha d}{\delta} \left( \frac{L}{2} \mu \sqrt{d} + G(\boldsymbol{\theta}_t) \right) + \mu \sqrt{d}$, we have $\nabla^2 \mathcal{L}(\boldsymbol{\theta}) \preceq H(\boldsymbol{\theta}_t)$.*

*2. The effective rank of $H(\boldsymbol{\theta}_t)$, i.e., $\frac{\mathrm{Tr}(H(\boldsymbol{\theta}_t))}{\|H(\boldsymbol{\theta}_t)\|_2}$ is at most $r$ where $r \ll d$.*

Note that for vanilla SGD with the limit case of $\mu \to 0$, the revised local effective rank condition (Assumption 4.2) just boils down to the original local effective rank condition (Assumption 4.1). The term $\mu \sqrt{d}$ is a by-product due to strictly positive smoothing parameter $\mu$.

Now, we are ready to state our main convergence theorem.

**Theorem 4.1** (Convergence of MeZO-A$^3$dam). *Under the conditions (C-1) $\sim$ (C-3), and Assumption 4.2 with the following parameter settings*

$$\alpha = \Theta \left( \frac{(1 - \beta_1)\delta}{\kappa_\delta L r} \right), \quad 1 - \beta_1 \leq \min\{1, c_1 \varepsilon^2\}, \quad T = \frac{1}{(1 - \beta_1)^2}, \quad \mu \lessapprox \frac{1}{d\sqrt{d}}$$

*where the constant $c_1$ denotes the small enough constant, ZO-Adam is guaranteed to converge to $\varepsilon$-stationary point with the complexity given by*

$$\mathbb{E}_R \left[ \|\nabla \mathcal{L}(\boldsymbol{\theta}_R)\|^2 \right] \leq \left( \frac{c_1 \Delta \kappa_\delta L r}{\beta_1} \left( 1 + \frac{\sqrt{\zeta}}{\delta} \right) + \frac{\kappa_\delta G^2}{\beta_1} + \frac{(\kappa_\delta + c_1)}{\beta_1 B} \sigma^2 \right) \mathcal{O}(\varepsilon^2) = \mathcal{O}(\varepsilon^2).$$

We make several remarks on our Theorem 4.1.

**On Novelty.** In fact, there were a study (Chen et al., 2019) on zeroth-order adaptive gradient methods. However, this work did not consider the exact Adam update rule. Thus, we emphasize that our analysis is the first attempt for the convergence guarantee of the zeroth-order version of Adam.

**On Dependency of $d$.** The complexity in Theorem 4.1 seems to be independent of the problem dimension $d$, however, the constants $\kappa_\delta$ and $\zeta$ can rely on $d$ implicitly. In this sense, the next proposition and corollary demonstrate the condition for completely dimension-free convergence rate.

**Proposition 4.1** (Condition on $\delta$ for Dimension-Free Convergence). *Given $\eta \in (0, 1)$, we assume that the parameter $\delta$ satisfies $\delta \geq \delta_0 \Omega \left( \sqrt{d \log (dT/\eta)} \right)$ where $\delta_0$ is the base adaptivity parameter of the first-order Adam. Then, with probability at least $1 - \eta$, we have $\kappa_\delta = \mathcal{O}(1)$ and $\sqrt{\zeta}/\delta = \mathcal{O}(1)$.*

**Corollary 4.1** (Dimension-Free Convergence Rate of MeZO-A$^3$dam). *Under the parameter settings in Theorem 4.1 and Proposition 4.1, ZO-Adam enjoys completely dimension-free convergence rate.*

**Remarks.** The Proposition 4.1 and Corollary 4.1 illustrate why the adaptivity parameter $\delta$ should be scaled in terms of convergence. It might be possible that one can improve the order of $\delta$ with respect to $d$ in Proposition 4.1 with the tighter bound. However, it is highly non-trivial in our experience. Note also that, in general, the total iteration $T$ is much smaller than the parameter dimension $d$ for fine-tuning LLMs (i.e. $d \gg T$). Despite the probabilistic guarantee in Proposition 4.1, the failure probability $\eta$ could be handled so that it does not significantly hurt the order of $\delta$. Therefore, it can be concluded that MeZO-A$^3$dam requires roughly at least $h(d) = \sqrt{d \log(d)}$ to ensure the dimension-free convergence rate. We will corroborate this remark through experiments in Section 5.

## 4.2 Generalization

Along with the convergence guarantee, we also provide the theoretical insights of MeZO-Adam in the perspective of the generalization. In pursuit of this, we use the uniform stability (Bousquet & Elisseeff, 2002; Hardt et al., 2016; Lei, 2023) of the randomized optimization algorithm (e.g. SGD).

For our arguments, we summarize the notations used in this section. We denote $\mathcal{A}$ by a randomized optimization algorithm such as SGD or Adam and $S \in \mathcal{Z}^n$ by the training dataset where $\mathcal{Z}^n$ represents the collection of datasets with the size $n$ sampled from the data distribution $\mathcal{D}$. The quantity $\mathcal{A}(S)$ represents the trained parameter using the algorithm $\mathcal{A}$ on the training dataset $S$.

Now, we start with the definition of the generalization error.

**Definition 4.1** (Generalization Error). *The generalization error $\epsilon_{\text{gen}}$ is defined by the gap between the population risk $R(\boldsymbol{\theta}) = \mathbb{E}_{\boldsymbol{z}\sim\mathcal{D}}\left[\mathcal{L}(\boldsymbol{\theta};\boldsymbol{z})\right]$ and the empirical risk $R_S(\boldsymbol{\theta}) = (1/n)\sum_{i=1}^{n}\mathcal{L}(\boldsymbol{\theta};\boldsymbol{z}_i)$ evaluated on the dataset $S = \{\boldsymbol{z}_1, \boldsymbol{z}_2, \cdots, \boldsymbol{z}_n\}$ as $\epsilon_{\text{gen}} = \mathbb{E}_{\boldsymbol{z}\sim\mathcal{D}}\left[R(\mathcal{A}(S)) - R_S(\mathcal{A}(S))\right]$.*

According to previous studies (Bousquet & Elisseeff, 2002; Hardt et al., 2016) on the generalization, $\epsilon_{\text{gen}}$ is closely related to the uniform stability of the optimization algorithm. Thus, we introduce the notion of the uniform stability.

**Definition 4.2** (Uniform Stability (Bousquet & Elisseeff, 2002; Hardt et al., 2016)). *The randomized algorithm $\mathcal{A}$ is said to be $\epsilon_{\text{stab}}$-uniformly stable if for all neighboring datasets $S, S' \in \mathcal{Z}^n$ such that $S$ and $S'$ differ in only one sample, we have $\mathbb{E}_{\mathcal{A},S}\left[\mathcal{L}(\mathcal{A}(S);\boldsymbol{z}) - \mathcal{L}(\mathcal{A}(S');\boldsymbol{z})\right] \leq \epsilon_{\text{stab}}$.*

The uniform stability measure how sensitive the algorithm $\mathcal{A}$ is for each training sample. The next lemma provides the connections between the generalization error and the uniform stability.

**Lemma 4.1** (Theorem 2.2 in Hardt et al. (2016)). *If $\mathcal{A}$ is $\epsilon_{\text{stab}}$-uniformly stable, then the generalization error is bounded by $|\epsilon_{\text{gen}}| \leq \epsilon_{\text{stab}}$.*

Thanks to Lemma 4.1, all we have to show is that MeZO-A$^3$dam is indeed uniformly stable with suitable stability bound. To this end, we assume standard conditions in this line of work.

**(G-1)** The loss function $\mathcal{L}_i = \mathcal{L}(\cdot, \boldsymbol{z}_i)$ for each datapoint $\boldsymbol{z}_i$ is $L$-smooth, i.e., $\|\nabla\mathcal{L}_i(\boldsymbol{\theta}) - \nabla\mathcal{L}_i(\boldsymbol{\theta}')\| \leq L\|\boldsymbol{\theta} - \boldsymbol{\theta}'\|$ for all $\boldsymbol{\theta}, \boldsymbol{\theta}' \in \mathbb{R}^d$.

**(G-2)** Each loss function $\mathcal{L}_i$ is $G$-Lipschitz, i.e., $|\mathcal{L}_i(\boldsymbol{\theta}) - \mathcal{L}_i(\boldsymbol{\theta}')| \leq G\|\boldsymbol{\theta} - \boldsymbol{\theta}'\|$ for all $\boldsymbol{\theta}, \boldsymbol{\theta}' \in \mathbb{R}^d$.

**(G-3)** Each loss $\mathcal{L}_i$ is bounded by strictly positive constant $M$, i.e., $|\mathcal{L}_i(\boldsymbol{\theta})| \leq M$.

**(G-4)** The minimum/maximum entry of the adaptation vector $\boldsymbol{v}_t$ and $\boldsymbol{v}_t'$ is uniformly bounded by

$$\gamma \leq \min_{i\in[d],t\in[T]}\{\boldsymbol{v}_{t,i}, \boldsymbol{v}_{t,i}'\}, \quad \zeta \geq \max_{i\in[d],t\in[T]}\{\boldsymbol{v}_{t,i}, \boldsymbol{v}_{t,i}'\}$$

where $\boldsymbol{v}_t'$ represent the preconditioner constructed during training on the dataset $S'$.

The conditions (G-1) $\sim$ (G-3) are standard in the uniform stability framework (Hardt et al., 2016; Guo et al., 2024) The last condition (G-4) is very similar to the condition (C-3) in convergence guarantee. This condition is also required for generalization analysis of first-order adaptive gradient methods (Nguyen et al., 2022).

Under the above conditions, MeZO-Adam is indeed uniformly stable by the following theorem.

**Theorem 4.2** (Uniform Stability of MeZO-A$^3$dam). *Under the conditions (G-1) $\sim$ (G-4) with the following parameter configurations*

$$\alpha_t = \frac{\alpha}{t}, \quad 1 - \beta_{1,t} = \frac{c_1}{nt}, \quad 1 - \beta_{2,t} = \frac{c_2}{dnt}, \quad \mu \lessapprox \frac{c_\mu}{d\sqrt{d}}$$

*where $\alpha, c_1, c_2$, and $c_\mu$ are constants, MeZO-Adam is uniformly stable with the stability bound as*

$$\epsilon_{\text{stab}} \leq \frac{1}{n}\left(2MQ_1 + Q_2\right)T^{1-\frac{1}{1+q}}$$

*for strictly positive constant $q > 0$. Note that the constants $Q_i$'s and $q$ satisfy $Q_i, q \propto \frac{d}{\delta^{3/2}}$.*

Note that the recent work (Liu et al., 2024a) derives the generalization error bound of various zeroth-order optimization algorithms under the uniform stability framework, however, it does not include the error bounds of any adaptive gradient methods. In terms of first-order optimization, Nguyen et al. (2022) discuss the uniform stability of the adaptive gradient methods including Adam. However, it only considers the case of $\beta_{1,t} = 0$, which boils down to the RMSprop. Therefore, in terms of both zeroth-order optimization and adaptive gradient methods, our Theorem 4.2 provides the first generalization analysis with exactly non-zero $\beta_1$ and $\beta_2$.

Similar to the convergence analysis, it can be also seen in Theorem 4.2 that the adaptivity parameter $\delta$ should be scaled to obtain dimension-free generalization error bound due to the constants $Q_i$'s.

**Corollary 4.2** (Dimension-Free Generalization Error Bound of MeZO-A$^3$dam). *Under the adaptivity parameter $\delta \geq \delta_0\Omega\left(d^{2/3}\right)$ where $\delta_0$ denotes the base adaptivity parameter of the first-order Adam, then we have dimension-free stability bound in Theorem 4.2 with $Q_i = \mathcal{O}(1)$ w.r.t. $d$ for $i \in \{1, 2\}$, which results in dimension-free generalization error bound as $\epsilon_{\text{gen}} \leq \mathcal{O}(\frac{T^{1-\frac{1}{1+q}}}{n})$ by Lemma 4.1.*

**Remarks.** According to Corollary 4.1 and Corollary 4.2, it is clear that the adaptivity parameter $\delta$ should be scaled to achieve the dimension-free bounds in terms of both convergence/generalization, which is strong theoretical evidence for our weak adaptivity hypothesis. In the next section, we will show that our hypothesis indeed empirically holds with an appropriate choice of scaling function $h$.

## 5 EXPERIMENTS

In this section, we evaluate the efficacy of MeZO-A³dam and validate our weak adaptivity hypothesis on fine-tuning LMs with various-sized models and tasks. In Section 5.1, we summarize the experimental setup and the main comparisons among different ZO methods will be presented in Section 5.2. In addition, we analyze the memory usage of different ZO methods and leverage an off-the-shelf low-bit optimizer for MeZO-A³dam, verifying its effectiveness in Section 5.3.

### 5.1 EXPERIMENTAL SETUP

In all experiments, we follow the experimental setup of MeZO-SVRG (Gautam et al., 2024) such as models, datasets, prompt-free format, and hyperparameters. The details on the experimental setup are provided in Appendix B.

**Models.** We use DistilBERT (Sanh, 2019) and RoBERTa-large (Liu, 2019) as representative masked language models, and GPT2-XL (Radford et al., 2019), OPT-2.7B, and OPT-6.7B (Zhang et al., 2022) as autoregressive language models. The language models are trained in single precision (FP32), while the large model (OPT-6.7B) is trained using half-precision (BF16).

**Datasets.** As downstream tasks, we consider SST-2 (Socher et al., 2013), MNLI (Williams et al., 2018), QNLI (Wang et al., 2019b), and CoLA (Warstadt, 2019) from the GLUE benchmark (Wang et al., 2019b). For the large-scale model, OPT-6.7B, we evaluate each algorithm on SST-2 and RTE (Wang et al., 2019b) from the GLUE benchmark, as well as BoolQ (Clark et al., 2019) and WiC (Pilehvar & Camacho-Collados, 2019) from the SuperGLUE benchmark (Wang et al., 2019a). For each task, we randomly sample 512/256 examples from the training/validation set, respectively.

**Empirical choices of scaling function $h$.** Based on our theory in Proposition 4.1 and Corollary 4.2, $\delta$ should be scaled by at least $\max\{\sqrt{d\log(d)}, d^{2/3}\} = d^{2/3}$ relative to the base adaptivity parameter $\delta_0 = 10^{-8}$. Given that our theoretical results provide lower bounds for $h$, we explore a range of approximately from $h(d) = 0.1d^{2/3}$ to $h(d) = 10d^{2/3}$. For instance, the adjusted adaptivity is approximately at least $1 \times 10^{-3}$ for models with fewer than 100M parameters, $5 \times 10^{-3}$ for models up to 500M, $1 \times 10^{-2}$ for models up to 2B parameters, and $3 \times 10^{-2}$ for models up to 7B parameters.

**Computational Resources.** All the experiments are conducted on a single GPU machine and the different GPUs are used according to the model sizes. We use NVIDIA RTX 2080 for the masked LMs (DistilBERT, RoBERTa-large) and NVIDIA RTX A6000 for the medium-sized autoregressive models (GPT2-XL, OPT-2.7B). In particular, for the large autoregressive model (OPT-6.7B), Intel Gaudi-2 (96GB) GPUs are used.

### 5.2 LANGUAGE MODEL FINE-TUNING PERFORMANCE

**Generalization.** Table 2 and 3 provide comprehensive comparisons of various ZO methods across a range of model sizes and tasks. Our proposed MeZO-A³dam demonstrates outstanding performance, superior to other baselines in both small and large-scale models.

In Table 2, MeZO-A³dam consistently outperforms MeZO, MeZO-SVRG, and MeZO-Adam by a great margin for DistilBERT and RoBERTa-large. Specifically, MeZO-A³dam achieves the highest average rank with MeZO-SVRG and MeZO-Adam showing similar rankings. In terms of performance, MeZO-A³dam achieves an average improvement of 36.6%, 6.4%, and 6.8% over MeZO, MeZO-SVRG, and MeZO-Adam, respectively. These results highlight the advantages of adaptivity adjustments, which provide strong empirical evidence for our weak adaptivity hypothesis.

Table 3 further illustrates the scalability of adaptivity adjustments when applied to larger models. MeZO-A³dam consistently meets or surpasses the performance of baselines across all datasets, achieving an average rank of 1.2. In particular, MeZO-A³dam improves the performance about 25.0%,

Table 2: Validation accuracy comparison among ZO methods for masked LMs on various datasets. Note that the results for MeZO and MeZO-SVRG are read off from Gautam et al. (2024).

| Method | DistilBERT (66M) | | | | RoBERTa-large (355M) | | | | Avg. Rank |
|---|---|---|---|---|---|---|---|---|---|
| | SST-2 | MNLI | QNLI | CoLA | SST-2 | MNLI | QNLI | CoLA | |
| MeZO (Malladi et al., 2023) | 52 | 36 | 50 | 63 | 56 | 43 | 59 | 68 | 4.0 |
| MeZO-SVRG (Gautam et al., 2024) | 72 | 46 | 68 | 68 | 84 | 49 | 80 | 79 | 2.6 |
| MeZO-Adam (Malladi et al., 2023) | 77 | 47 | 71 | **69** | 85 | 48 | 71 | 75 | 2.3 |
| **MeZO-A$^3$dam (Ours)** | **81** | **53** | **72** | **69** | **88** | **52** | **82** | **81** | **1.0** |

Table 3: Validation accuracy comparison among ZO methods for autoregressive LMs on various datasets. Note that the results for MeZO and MeZO-SVRG are read off from Gautam et al. (2024). The mark $\dagger$ represents our reproduced results based on the official implementation.[1]

| Method | GPT2-XL (1.5B) | | | OPT-2.7B | | | OPT-6.7B | | | | Avg. Rank |
|---|---|---|---|---|---|---|---|---|---|---|---|
| | SST-2 | MNLI | CoLA | SST-2 | MNLI | CoLA | SST-2 | RTE | BoolQ | WiC | |
| MeZO (Malladi et al., 2023) | 59 | 41 | 61 | 61 | 42 | 62 | 74 | 56 | 65 | 52 | 3.8 |
| MeZO-SVRG (Gautam et al., 2024) | 65 | 44$^\dagger$ | 69 | 65 | 43$^\dagger$ | 67 | 77 | 59 | 65$^\dagger$ | 59 | 2.7 |
| MeZO-Adam (Malladi et al., 2023) | 81 | **48** | **73** | 86 | 45 | **74** | 92 | 59 | 65 | 59 | 1.6 |
| **MeZO-A$^3$dam (Ours)** | **89** | 47 | **73** | **92** | **57** | 73 | **92** | **63** | **68** | **62** | **1.2** |

16.9%, and 5.7% on average compared to MeZO, MeZO-SVRG, and MeZO-Adam, respectively. These results further emphasize its superiority for larger models and more complex tasks.

**Convergence.** Figure 2 compares the GPU hours required by various ZO methods to attain equivalent performance levels. Notably, our MeZO-A$^3$dam consistently achieves significantly faster convergence times than MeZO-SVRG and MeZO-Adam, underscoring its superior computational efficiency.

In Figure 2(a), MeZO-A$^3$dam significantly reduces the GPU hours required to achieve the same performance level as MeZO-Adam. On average, MeZO-A$^3$dam delivers computation times that are $\times 1.8$ times faster than MeZO-Adam across all models and datasets. Specifically, MeZO-A$^3$dam achieves $\times 3.0$, $\times 2.5$, and $\times 1.7$ faster convergence than MeZO-Adam for OPT-2.7B on MNLI, OPT-6.7B on BoolQ, and GPT2-XL on SST-2, respectively. Our results confirm the clear advantage of adaptivity adjustment in enhancing both convergence speed and scalability.

Similarly, the comparison of the convergence time between MeZO-A$^3$dam and MeZO-SVRG in Figure 2(b) shows a drastic reduction in the convergence time. MeZO-A$^3$dam provides an average of $\times 12.6$ faster convergence speed compared to MeZO-SVRG. For instance, the same level of performance as MeZO-SVRG can be obtained by MeZO-A$^3$dam $\times 50.3$, $\times 25.1$, and $\times 9.8$ faster for GPT2-XL on CoLA, OPT-2.7B on CoLA, and OPT-6.7B on BoolQ. These substantial reductions in computational time across models and tasks illustrate that MeZO-A$^3$dam offers a faster convergence, making it an excellent choice for training large-scale models in a time-efficient manner.

## 5.3 MEMORY CONSUMPTION ANALYSIS AND COMPRESSED OPTIMIZER

In terms of memory consumption, MeZO-A$^3$dam requires at least $3d$ memory, where each $d$ counts to the dimension of $\boldsymbol{\theta}_t$, $\boldsymbol{m}_t$, and $\boldsymbol{v}_t$ in Alg. 1. This memory requirement might appear inconsistent with memory efficiency, which is the central goal of ZO optimization. However, it is important to note that adaptive gradient methods such as Adam are widely used in fine-tuning LLMs. Thus, extensive studies have been conducted to reduce memory consumption. In particular, we leverage the well-established memory-efficient solution, 8-bit Adam (Dettmers et al., 2022), which substantially reduces the memory footprint of MeZO-A$^3$dam to $1.5d$ for 32-bit training and $2d$ for 16-bit training.

As illustrated in Figure 3(a), MeZO-A$^3$dam with the 8-bit optimizer demonstrates a substantial reduction in memory usage compared to both MeZO-SVRG and the standard 32-bit MeZO-A$^3$dam. Specifically, the 8-bit MeZO-A$^3$dam requires, on average, 40.3% and 43.4% less memory than MeZO-SVRG and 32-bit MeZO-A$^3$dam respectively, although it demands 33.5% more memory than MeZO. Moreover, the 8-bit MeZO-A$^3$dam enables fine-tuning of the OPT-13B model on a single 80GB GPU, which would not be feasible with either MeZO-SVRG or 32-bit MeZO-A$^3$dam. Furthermore, as shown in Figure 3(b), both the 8-bit and 32-bit variants of MeZO-A$^3$dam yield

---

[1]We attempted to reproduce all the results of MeZO-SVRG; unfortunately, for some cases, replication was not feasible. In such cases, we report the results obtained by running the official implementation ourselves.

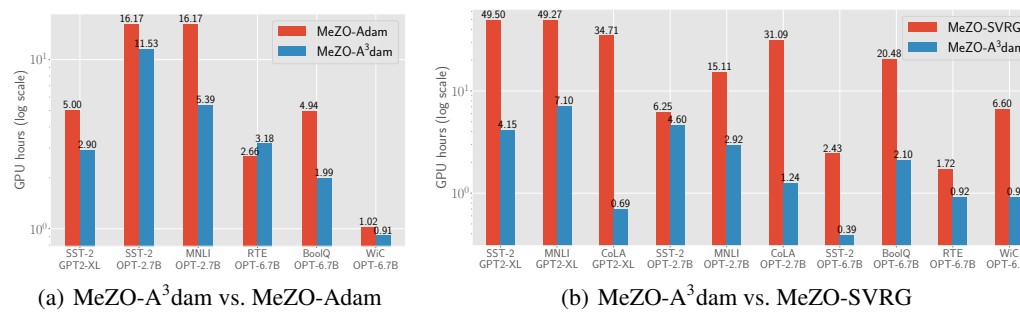

(a) MeZO-A³dam vs. MeZO-Adam  (b) MeZO-A³dam vs. MeZO-SVRG

Figure 2: Required GPU hours to achieve equivalent performance levels for MeZO-A³dam and two different methods across various models and tasks. The results are shown only where MeZO-A³dam provides better generalization than the other one.

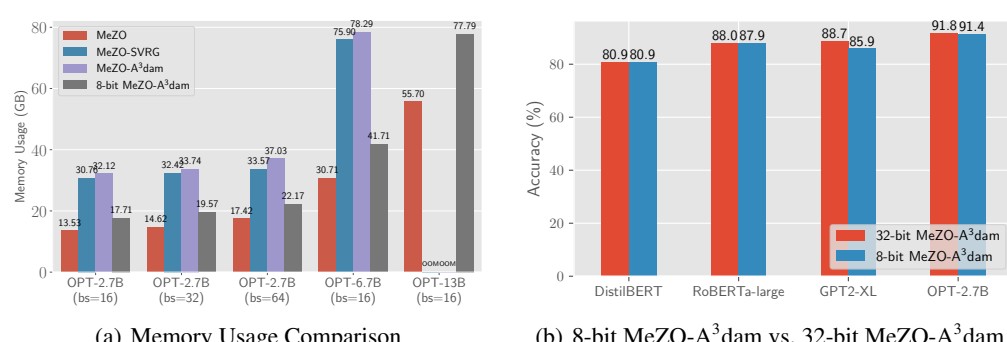

(a) Memory Usage Comparison  (b) 8-bit MeZO-A³dam vs. 32-bit MeZO-A³dam

Figure 3: **(a)** Memory usage comparison across methods in GB for different sizes of OPT models and batch sizes in single precision (FP32) training. **(b)** validation accuracy comparisons for 32-bit and 8-bit MeZO-A³dam on SST-2 and various models.

comparable performance across all considered models, thereby demonstrating that the 8-bit optimizer can be employed in MeZO-A³dam without sacrificing performance.

Using the 8-bit Adam, we dramatically reduce the memory requirement of MeZO-A³dam, bringing it close to that of inference and making it a highly competitive solution for ZO fine-tuning of language models. This allows MeZO-A³dam to maintain its advantages of adaptive gradients while achieving memory efficiency, possibly positioning it as an optimal choice. It is noteworthy that, although the theoretical memory requirements for both MeZO-SVRG and MeZO-A³dam are identical (at least $3d$), our MeZO-A³dam can leverage off-the-shelf memory-efficient optimizers such as the 8-bit Adam, which is not available for MeZO-SVRG. This distinction reinforces MeZO-A³dam as a more favorable option regarding both convergence speed and memory efficiency in fine-tuning LLMs.

# 6 CONCLUSION

In this paper, we revisited MeZO-Adam and highlighted the advantage of using adaptive gradients in zeroth-order fine-tuning in a prompt-free scenario. Further, we provided an important observation that reducing the level of adaptivity of MeZO-Adam is highly recommended in the zeroth-order regime, which is hypothesized as *weak adaptivity hypothesis*. Given our hypothesis, we proposed a MeZO-A³dam, which adjusts the adaptivity according to the parameter dimension. We analyzed the convergence and generalization of MeZO-A³dam, providing dimension-free guarantees and presenting strong theoretical evidence for our weak adaptivity hypothesis. We also validated that MeZO-A³dam outperforms several existing ZO baselines in practice for fine-tuning various sizes of language models on benchmark tasks across all three aspects: generalization, convergence, and memory consumption, which empirically corroborated the weak adaptivity hypothesis. In future work, we plan to investigate the zeroth-order adaptive gradient methods for more challenging loss landscape.

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

# APPENDIX

## A    RELATED WORK

**Fine-tuning Language Models.**    One popular approach is *parameter-efficient fine-tuning* (PEFT), where only a small subset of parameters are optimized. Examples include Low-Rank Adaptation (LoRA) (Hu et al., 2022), prefix-tuning (Li & Liang, 2021), and prompt-tuning (Lester et al., 2021). While PEFT reduces the number of parameters involved in optimization, it still requires significant memory and computation due to the need for backpropagation and the storage of intermediate activations, optimizer states, and gradients. Moreover, obtaining significant performance gains often necessitates expanding the number of parameters being fine-tuned.

Another method for adapting LLMs is *in-context learning* (ICL) (Min et al., 2022), which provides task-specific instructions and examples to leverage LLMs' inherent language understanding. While gradient-free, ICL faces challenges such as selecting proper instructions and examples, increased inference times due to long context lengths, and lower performance compared to fine-tuning.

**Fine-tuning LMs with Zeroth-Order (ZO) Optimization.**    Zeroth-order optimization, long explored in conventional machine learning (Spall, 1992; Ghadimi & Lan, 2013), was first applied to LLM fine-tuning by MeZO (Malladi et al., 2023). This method reduces memory consumption by generating perturbations on-the-fly through random seeds, eliminating the need to store large perturbation vectors. As a result, ZO optimization requires only inference-level resources. MeZO also provides theoretical guarantees, suggesting that scaling the learning rate by the problem dimension ensures dimension-free convergence. However, this leads to slower convergence rates. Additionally, MeZO's use of hand-crafted task-specific prompts introduces extra overhead, limiting its general applicability.

MeZO-SVRG (Gautam et al., 2024) improves zeroth-order optimization by incorporating the SVRG algorithm, addressing the challenges of non-prompted fine-tuning through reduced gradient variance, which leads to faster convergence and better performance. Similarly, SparseMeZO (Liu et al., 2024b) leverages sparsity by updating only a subset of parameters for quicker convergence, while Fisher-informed sparsity (Guo et al., 2024) selects parameters based on Fisher information for greater efficiency. Additionally, Zhang et al. (2024) offers a comprehensive benchmark of ZO optimization methods like SGD, SignSGD, and Adam across various models and tasks, with results suggesting that while Adam may not always outperform SGD, its success in LLM fine-tuning merits further exploration.

## B    EXPERIMENTAL SETUP

In all experiments, we adopted the same experimental setup as used in MeZO-SVRG (Gautam et al., 2024) including models, datasets, a prompt-free approach, and hyperparameters.

### B.1    DATASETS

We focus on fine-tuning classification tasks in our experiments. Specifically, we utilize datasets from the General Language Understanding Evaluation (GLUE) (Wang et al., 2019b) benchmark, such as Stanford Sentiment Treebank (SST-2) (Socher et al., 2013) for sentiment analysis, Multi-Genre Natural Language Inference (MNLI) (Williams et al., 2018), Question Natural Language Inference (Wang et al., 2019b), and the Corpus of Linguistic Acceptability (CoLA) (Warstadt, 2019). Additionally, we extend the evaluation to larger models like OPT-6.7B by testing them on more complex tasks from the SuperGLUE (Wang et al., 2019a) benchmark, including Recognizing Textual Entailment (RTE) (Wang et al., 2019b), BoolQ (Clark et al., 2019), and Word-in-Context (WiC) (Pilehvar & Camacho-Collados, 2019).

The datasets are sourced from the Huggingface `datasets` library. For each dataset, we randomly select 512 samples for training and 256 for validation, reporting validation accuracy since the test labels for GLUE and SuperGLUE benchmarks are unavailable. This setup is identical to that used in MeZO-SVRG (Gautam et al., 2024).

### B.2    MODELING AND IMPLEMENTATION

We utilize DistilBERT (Sanh, 2019) and RoBERTa-large (Liu, 2019) as representative masked language models, alongside GPT2-XL (Radford et al., 2019), OPT-2.7B, and OPT-6.7B (Zhang et al., 2022) as autoregressive models. The small and medium-sized models (DistilBERT, RoBERTa-large, GPT2-XL, and OPT-2.7B) are trained using single precision (FP32), while the larger OPT-6.7B is trained using half-precision (BF16).

For the experiments, we rely on the Huggingface `transformers` library to implement the models. Since we focus on classification tasks, we employ models from the `AutoModelForSequenceClassification`

Table 4: The hyperparameter search grid for the DistilBERT (Sanh, 2019) experiments. We do not use any learning rate scheduling for MeZO-Adam and MeZO-A$^3$dam. The final results are produced using the configuration indicated by the bold values in the grid.

| Algorithm | Hyperparameters | Values |
|---|---|---|
| MeZO-Adam | Batch size | $\{32, \mathbf{64}\} \times$ |
| | Learning rate | $\{\mathbf{1 \times 10^{-4}}, 5 \times 10^{-5}, 1 \times 10^{-5}\} \times$ |
| | $\delta$ | $\{\mathbf{1 \times 10^{-8}}\} \times$ |
| | Total Steps | $\{\mathbf{200K}\}$ |
| MeZO-A$^3$dam | Batch size | $\{32, \mathbf{64}\} \times$ |
| | Learning rate | $\{\mathbf{1 \times 10^{-4}}, 5 \times 10^{-5}, 1 \times 10^{-5}\} \times$ |
| | $\delta$ | $\{1 \times 10^{-4}, \mathbf{1 \times 10^{-3}}, 1 \times 10^{-2}\} \times$ |
| | Total Steps | $\{\mathbf{200K}\}$ |

Table 5: The hyperparameter search grid for the RoBERTa-large (Liu, 2019) experiments. We do not use any learning rate scheduling for MeZO-Adam and MeZO-A$^3$dam. The final results are produced using the configuration indicated by the bold values in the grid.

| Algorithm | Hyperparameters | Values |
|---|---|---|
| MeZO-Adam | Batch size | $\{32, \mathbf{64}\} \times$ |
| | Learning rate | $\{1 \times 10^{-4}, \mathbf{5 \times 10^{-5}}, 1 \times 10^{-5}\} \times$ |
| | $\delta$ | $\{\mathbf{1 \times 10^{-8}}\} \times$ |
| | Total Steps | $\{\mathbf{96K}\}$ |
| MeZO-A$^3$dam | Batch size | $\{32, \mathbf{64}\} \times$ |
| | Learning rate | $\{1 \times 10^{-4}, \mathbf{5 \times 10^{-5}}, 1 \times 10^{-5}\} \times$ |
| | $\delta$ | $\{5 \times 10^{-4}, \mathbf{5 \times 10^{-3}}, 5 \times 10^{-2}\} \times$ |
| | Total Steps | $\{\mathbf{96K}\}$ |

and `OPTModelForSequenceClassification` classes, which add a classification head to the pre-trained models.

The specific pre-trained models used in the experiments are: `distilbert-base-cased` for Distil-BERT (Sanh, 2019), `roberta-large` for RoBERTa-large (Liu, 2019), `openai-community/gpt2-xl` for GPT2-XL (Radford et al., 2019), and `facebook/opt-2.7b` and `facebook/opt-6.7b` for OPT-2.7B and OPT-6.7B (Zhang et al., 2022), respectively.

### B.3 HYPERPARAMETERS

Table 4 $\sim$ 8 provide the hyperparameter search grid used in our experiments. For reproducing the results in MeZO-SVRG, we follow to the hyperparameters reported in MeZO-SVRG (Gautam et al., 2024). It is important to note that MeZO-SVRG increases the total training steps by four times for batch size 64 and by three times for batch size 128 to match the total number of queries (i.e., one forward pass for a single sample). However, in our MeZO-Adam and MeZO-A$^3$dam experiments for autoregressive LMs, we maintained the same total steps as in MeZO-SVRG without further increasing them based on number of queries.

Table 6: The hyperparameter search grid for the GPT2-XL ([Radford et al., 2019]) experiments. We do not use any learning rate scheduling for MeZO-Adam and MeZO-A$^3$dam. The final results are produced using the configuration indicated by the bold values in the grid.

| Algorithm | Hyperparameters | Values |
|---|---|---|
| MeZO-Adam | Batch size | $\{32, \mathbf{64}\} \times$ |
| | Learning rate | $\{1 \times 10^{-4}, \mathbf{2 \times 10^{-4}}, 5 \times 10^{-4}\} \times$ |
| | $\delta$ | $\{\mathbf{1 \times 10^{-8}}\} \times$ |
| | Total Steps | $\{\mathbf{8K}\}$ |
| MeZO-A$^3$dam | Batch size | $\{32, \mathbf{64}\} \times$ |
| | Learning rate | $\{1 \times 10^{-4}, \mathbf{2 \times 10^{-4}}, 5 \times 10^{-4}\} \times$ |
| | $\delta$ | $\{1 \times 10^{-3}, \mathbf{1 \times 10^{-2}}, 1 \times 10^{-1}\} \times$ |
| | Total Steps | $\{\mathbf{8K}\}$ |

Table 7: The hyperparameter search grid for the OPT-2.7B ([Zhang et al., 2022]) experiments. We do not use any learning rate scheduling for MeZO-Adam and MeZO-A$^3$dam. The final results are produced using the configuration indicated by the bold values in the grid.

| Algorithm | Hyperparameters | Values |
|---|---|---|
| MeZO-Adam | Batch size | $\{32, \mathbf{64}\} \times$ |
| | Learning rate | $\{1 \times 10^{-5}, 2 \times 10^{-5}, \mathbf{5 \times 10^{-5}}\} \times$ |
| | $\delta$ | $\{\mathbf{1 \times 10^{-8}}\} \times$ |
| | Total Steps | $\{\mathbf{8K}\}$ |
| MeZO-A$^3$dam | Batch size | $\{32, \mathbf{64}\} \times$ |
| | Learning rate | $\{1 \times 10^{-5}, 2 \times 10^{-5}, \mathbf{5 \times 10^{-5}}\} \times$ |
| | $\delta$ | $\{1 \times 10^{-3}, \mathbf{1 \times 10^{-2}}, 1 \times 10^{-1}\} \times$ |
| | Total Steps | $\{\mathbf{8K}\}$ |

Table 8: The hyperparameter search grid for the OPT-6.7B ([Zhang et al., 2022]) experiments. We do not use any learning rate scheduling for MeZO-Adam and MeZO-A$^3$dam. The final results are produced using the configuration indicated by the bold values in the grid.

| Algorithm | Hyperparameters | Values |
|---|---|---|
| MeZO-Adam | Batch size | $\{\mathbf{128}\} \times$ |
| | Learning rate | $\{1 \times 10^{-5}, 2 \times 10^{-5}, \mathbf{5 \times 10^{-5}}\} \times$ |
| | $\delta$ | $\{\mathbf{1 \times 10^{-8}}\} \times$ |
| | Total Steps | $\{\mathbf{8K}\}$ |
| MeZO-A$^3$dam | Batch size | $\{\mathbf{128}\} \times$ |
| | Learning rate | $\{1 \times 10^{-5}, 2 \times 10^{-5}, \mathbf{5 \times 10^{-5}}\} \times$ |
| | $\delta$ | $\{1 \times 10^{-3}, 1 \times 10^{-2}, \mathbf{1 \times 10^{-1}}\} \times$ |
| | Total Steps | $\{\mathbf{8K}\}$ |

## C    PROOF OF THEOREM 4.1

Recall the Adam update rule, which would be

$$\widehat{\boldsymbol{g}}_t = \widehat{\nabla}\mathcal{L}(\boldsymbol{\theta}_t; \mathcal{B}_t)$$
$$\boldsymbol{m}_t = \beta_1 \boldsymbol{m}_{t-1} + (1 - \beta_1)\widehat{\boldsymbol{g}}_t$$
$$\boldsymbol{V}_t = \beta_2 \boldsymbol{V}_{t-1} + (1 - \beta_2)\widehat{\boldsymbol{g}}_t^{\odot 2}$$
$$\widehat{\boldsymbol{m}}_t = \frac{\boldsymbol{m}_t}{1 - \beta_1^t}$$
$$\widehat{\boldsymbol{V}}_t = \frac{\boldsymbol{V}_t}{1 - \beta_2^t}$$
$$\boldsymbol{\theta}_{t+1} = \boldsymbol{\theta}_t - \alpha(\widehat{\boldsymbol{V}}_t^{1/2} + \delta\boldsymbol{I}_d)^{-1}\widehat{\boldsymbol{m}}_t$$

We make several auxiliary lemmas for our analysis.

**Lemma C.1** (Norm of Gaussian Vector). *For a given $\delta \in (0, 1/2)$, with probability at least $1 - \delta$, the norm of Gaussian random vector $\boldsymbol{u} \sim \mathcal{N}(\boldsymbol{0}, \boldsymbol{I}_d)$ is bounded as*

$$\|\boldsymbol{u}\| \leq \sqrt{2d\log(1/\delta)}$$

*Proof.* From the concentration bound, we have

$$\mathbb{P}\left[\|\boldsymbol{u}\| > \xi\right] \leq 2\exp\left(-\frac{\xi^2}{2d}\right)$$

Hence, we have

$$\mathbb{P}[\|\boldsymbol{u}\| \leq \xi] \geq 1 - 2\exp\left(-\frac{\xi^2}{2d}\right)$$

Let $\delta = 2\exp(-\xi^2/2d)$. Then, we have

$$\xi = \sqrt{2d\log\left(\frac{2}{\delta}\right)}$$

$\square$

**Lemma C.2.** *For positive semi-definite matrices $A, B \in \mathbb{R}^{d \times d}$, the following statement holds.*

$$\mathrm{Tr}(AB) \leq \mathrm{Tr}(A)\mathrm{Tr}(B)$$

*Proof.* Let $\{v_i\}_{i=1}^d$ be an orthonormal basis for $B$ and $\{\lambda_i\}_{i=1}^d$ be corresponding eigenvalues. Then, we have

$$\mathrm{Tr}(AB) = \sum_{i=1}^d \langle ABv_i, v_i \rangle = \sum_{i=1}^d \lambda_i \langle Av_i, v_i \rangle = \max_{i \in [d]} \lambda_i \mathrm{Tr}(A) \leq \mathrm{Tr}(A)\mathrm{Tr}(B).$$

$\square$

**Lemma C.3** (Nesterov & Spokoiny (2017)). *Let $\mu \sim \mathcal{N}(\boldsymbol{0}, \boldsymbol{I}_d)$. Then, the expectation of the moment satisfies*

$$\mathbb{E}\big[\|\boldsymbol{u}\|\big] = 1$$
$$\mathbb{E}\big[\|\boldsymbol{u}\|^2\big] = d$$
$$\mathbb{E}\big[\|\boldsymbol{u}\|^n\big] \leq (d + n)^{n/2}$$

*for $n \geq 2$.*

**Lemma C.4** (n-th Momentum of Quadratic Forms for $n = 1, 2,$ and 4 (Magnus et al., 1978)). *Let $\boldsymbol{u} \sim \mathcal{N}(\boldsymbol{0}, \boldsymbol{I}_d)$ and $A$ be a positive semi-definite matrix. Then, the expectation of the followings are*

$$\mathbb{E}\left[\boldsymbol{u}^\top A\boldsymbol{u}\right] = \mathrm{Tr}(A)$$

$$\mathbb{E}\left[(\boldsymbol{u}^\top A\boldsymbol{u})^2\right] = (\mathrm{Tr}(A))^2 + 2\mathrm{Tr}(A^2) \leq 3\mathrm{Tr}(A)^2$$

$$\mathbb{E}\left[(\boldsymbol{u}^\top A\boldsymbol{u})^4\right] = (\mathrm{Tr}(A))^4 + 32\mathrm{Tr}(A)\mathrm{Tr}(A^3) + 12\left(\mathrm{Tr}(A^2)\right)^2 + 12\left(\mathrm{Tr}(A)\right)^2\mathrm{Tr}(A^2) + 48\mathrm{Tr}(A^4)$$

$$\leq 105\mathrm{Tr}(A)^4$$

*The inequalities come from Lemma C.2.*

**Lemma C.5.** *For any $s, t \in [T]$, we have*

$$\mathbb{E}\left[\widehat{\boldsymbol{g}}_s^\mathsf{T} \boldsymbol{H}(\boldsymbol{\theta}_t)\widehat{\boldsymbol{g}}_s\right] \leq \frac{\sqrt{6}}{2}\mu^2 L^3 r(d+8)^2 + 6\sqrt{2}Lr\|\nabla\mathcal{L}(\boldsymbol{\theta}_t; \mathcal{B}_t)\|^2$$

$$\leq 2\mu^2 L^3 r(d+8)^2 + 10Lr\left(\|\nabla\mathcal{L}(\boldsymbol{\theta}_s)\|^2 + \frac{\sigma^2}{B}\right)$$

*Proof.* Recall the definition of $\widehat{\boldsymbol{g}}_s$ as

$$\widehat{\boldsymbol{g}}_s = \frac{\mathcal{L}(\boldsymbol{\theta}_s + \mu\boldsymbol{u}_s; \mathcal{B}_s) - \mathcal{L}(\boldsymbol{\theta}_s - \mu\boldsymbol{u}_s; \mathcal{B}_s)}{2\mu}\boldsymbol{u}_s$$

The expectation of the quadratic form can be computed as

$$\mathbb{E}\left[\widehat{\boldsymbol{g}}_s^\mathsf{T} \boldsymbol{H}(\boldsymbol{\theta}_t)\widehat{\boldsymbol{g}}_s\right] = \frac{1}{4\mu^2}\mathbb{E}\left[[\mathcal{L}(\boldsymbol{\theta}_s + \mu\boldsymbol{u}_s; \mathcal{B}_s) - \mathcal{L}(\boldsymbol{\theta}_s - \mu\boldsymbol{u}_s; \mathcal{B}_s)]^2\boldsymbol{u}_s^\mathsf{T}\boldsymbol{H}(\boldsymbol{\theta}_t)\boldsymbol{u}_s\right]$$

$$\leq \frac{1}{4\mu^2}\sqrt{\mathbb{E}\left[(\mathcal{L}(\boldsymbol{\theta}_s + \mu\boldsymbol{u}_s; \mathcal{B}_s) - \mathcal{L}(\boldsymbol{\theta}_s - \mu\boldsymbol{u}_s; \mathcal{B}_s))^4\right]}\sqrt{\mathbb{E}\left[\left(\boldsymbol{u}_s^\mathsf{T}\boldsymbol{H}(\boldsymbol{\theta}_t)\boldsymbol{u}_s\right)^2\right]}$$

Hence, we have

$$\mathcal{L}(\boldsymbol{\theta}_s + \mu\boldsymbol{u}_s; \mathcal{B}_s) - \mathcal{L}(\boldsymbol{\theta}_s - \mu\boldsymbol{u}_s; \mathcal{B}_s) \leq \mu^2 L\|\boldsymbol{u}_s\|^2 + 2\mu\langle\nabla\mathcal{L}(\boldsymbol{\theta}_s; \mathcal{B}_s), \boldsymbol{u}_s\rangle$$

Therefore, we have by Young's inequality

$$\left(\mathcal{L}(\boldsymbol{\theta}_s + \mu\boldsymbol{u}_s; \mathcal{B}_s) - \mathcal{L}(\boldsymbol{\theta}_s - \mu\boldsymbol{u}_s; \mathcal{B}_s)\right)^4 \leq 8\mu^8 L^4\|\boldsymbol{u}_s\|^8 + 128\mu^4\langle\nabla\mathcal{L}(\boldsymbol{\theta}_s; \mathcal{B}_s), \boldsymbol{u}_s\rangle^4$$

Hence, the expectation is

$$\mathbb{E}\left[\left(\mathcal{L}(\boldsymbol{\theta}_s + \mu\boldsymbol{u}_s; \mathcal{B}_s) - \mathcal{L}(\boldsymbol{\theta}_s - \mu\boldsymbol{u}_s; \mathcal{B}_s)\right)^4\right] \leq \mathbb{E}\left[8\mu^8 L^4\|\boldsymbol{u}_s\|^8 + 128\mu^4\langle\nabla\mathcal{L}(\boldsymbol{\theta}_s; \mathcal{B}_s), \boldsymbol{u}_s\rangle^4\right]$$

$$\leq 8\mu^8 L^4(d+8)^4 + 384\mu^4\|\nabla\mathcal{L}(\boldsymbol{\theta}_s; \mathcal{B}_s)\|^4$$

Finally, we arrive at

$$\sqrt{\mathbb{E}\left[\left(\mathcal{L}(\boldsymbol{\theta}_s + \mu\boldsymbol{u}_s; \mathcal{B}_s) - \mathcal{L}(\boldsymbol{\theta}_s - \mu\boldsymbol{u}_s; \mathcal{B}_s)\right)^4\right]} \leq \sqrt{8\mu^8 L^4(d+8)^4 + 384\mu^4\|\nabla\mathcal{L}(\boldsymbol{\theta}_s; \mathcal{B}_s)\|^4}$$

$$\leq \sqrt{8\mu^8 L^4(d+8)^4} + \sqrt{384\mu^4\|\nabla\mathcal{L}(\boldsymbol{\theta}_s; \mathcal{B}_s)\|^4}$$

$$\leq 2\sqrt{2}\mu^4 L^2(d+8)^2 + 8\sqrt{6}\mu^2\|\nabla\mathcal{L}(\boldsymbol{\theta}_s; \mathcal{B}_s)\|^2$$

The expectation of the quadratic form then would be

$$\mathbb{E}\left[\widehat{\boldsymbol{g}}_s^\mathsf{T} \boldsymbol{H}(\boldsymbol{\theta}_t)\widehat{\boldsymbol{g}}_s\right] \leq \frac{\sqrt{3}Lr}{4\mu^2}\left(2\sqrt{2}\mu^4 L^2(d+8)^2 + 8\sqrt{6}\mu^2\|\nabla\mathcal{L}(\boldsymbol{\theta}_s; \mathcal{B}_s)\|^2\right)$$

$$= \frac{\sqrt{6}}{2}\mu^2 L^3 r(d+8)^2 + 6\sqrt{2}Lr\|\nabla\mathcal{L}(\boldsymbol{\theta}_s; \mathcal{B}_s)\|^2$$

$$\leq 2\mu^2 L^3 r(d+8)^2 + 10Lr\|\nabla\mathcal{L}(\boldsymbol{\theta}_s; \mathcal{B}_s)\|^2$$

$$\leq 2\mu^2 L^3 r(d+8)^2 + 10Lr\left(\|\nabla\mathcal{L}(\boldsymbol{\theta}_s)\|^2 + \frac{\sigma^2}{B}\right)$$

$\square$

Note that the Adam update rule could be re-written as

$$(1 - \beta_1^t)\widehat{\boldsymbol{m}}_t = \beta_1(1 - \beta_1^{t-1})\widehat{\boldsymbol{m}}_{t-1} + (1 - \beta_1)\widehat{\boldsymbol{g}}_t$$

Hence, we have

$$\widehat{\boldsymbol{m}}_t = \frac{\beta_1(1 - \beta_1^{t-1})}{1 - \beta_1^t}\widehat{\boldsymbol{m}}_{t-1} + \frac{1 - \beta_1}{1 - \beta_1^t}\widehat{\boldsymbol{g}}_t$$

The sum of coefficient is

$$\frac{\beta_1 - \beta_1^t + 1 - \beta_1}{1 - \beta_1^t} = 1$$

Therefore, we could say

$$\widehat{\boldsymbol{m}}_t = c_t \widehat{\boldsymbol{m}}_{t-1} + (1 - c_t)\widehat{\boldsymbol{g}}_t$$

with initial condition $\widehat{\boldsymbol{m}}_1 = \widehat{\boldsymbol{g}}_1$ and $c_t = \frac{\beta_1(1-\beta_1^{t-1})}{1-\beta_1^t}$. Now, we define

$$\boldsymbol{\epsilon}_t = \widehat{\boldsymbol{m}}_t - \nabla\mathcal{L}_{\boldsymbol{u}}(\boldsymbol{\theta}_t)$$
$$\boldsymbol{\xi}_{t-1} = c_t(\boldsymbol{\epsilon}_{t-1} + \nabla\mathcal{L}_{\boldsymbol{u}}(\boldsymbol{\theta}_{t-1}) - \nabla\mathcal{L}_{\boldsymbol{u}}(\boldsymbol{\theta}_t))$$

We compute

$$\boldsymbol{\epsilon}_t = c_t(\boldsymbol{\epsilon}_{t-1} + \nabla\mathcal{L}_{\boldsymbol{u}}(\boldsymbol{\theta}_{t-1}) - \nabla\mathcal{L}_{\boldsymbol{u}}(\boldsymbol{\theta}_t)) + (1-c_t)(\widehat{\boldsymbol{g}}_t - \nabla\mathcal{L}_{\boldsymbol{u}}(\boldsymbol{\theta}_t))$$
$$= \boldsymbol{\xi}_{t-1} + (1-c_t)(\widehat{\boldsymbol{g}}_t - \nabla\mathcal{L}_{\boldsymbol{u}}(\boldsymbol{\theta}_t))$$

**Lemma C.6.** *For $d$-dimensional standard Gaussian vector $\boldsymbol{u} = (u_1, \cdots, u_d) \sim \mathcal{N}(\boldsymbol{0}, \boldsymbol{I}_d)$ and any vector $a = (a_1, \cdots, a_d)$, we have*

$$\mathbb{E}_{\boldsymbol{u}}\left[\|\langle\boldsymbol{a}, \boldsymbol{u}\rangle\boldsymbol{u}\|\right] \leq \sqrt{3}\|\boldsymbol{a}\|$$
$$\mathbb{E}_{\boldsymbol{u}}\left[\|\langle\boldsymbol{a}, \boldsymbol{u}\rangle\boldsymbol{u}\|^2\right] = 3\|\boldsymbol{a}\|^2$$

*Proof.* We directly compute the quantity to be expected as

$$\langle\boldsymbol{a}, \boldsymbol{u}\rangle\boldsymbol{u} = (a_1 u_1 + \cdots + a_d u_d)\boldsymbol{u}$$

$$= \left(\sum_{i=1}^{d} a_i u_i u_1, \sum_{i=1}^{d} a_i u_i u_2, \cdots, \sum_{i=1}^{d} a_i u_i u_d\right)$$

Thus, the expectation of the norm would be

$$\mathbb{E}\left[\|\langle\boldsymbol{a}, \boldsymbol{u}\rangle\boldsymbol{u}\|\right] = \mathbb{E}\left[\sqrt{\sum_{j=1}^{d}\left(\sum_{i=1}^{d} a_i u_i u_j\right)^2}\right]$$

$$\leq \sqrt{\mathbb{E}\left[\sum_{j=1}^{d}\left(\sum_{i=1}^{d} a_i u_i u_j\right)^2\right]}$$

$$= \sqrt{\sum_{j=1}^{d}\mathbb{E}\left[\left(\sum_{i=1}^{d} a_i u_i u_j\right)^2\right]}$$

The first inequality comes from Jensen's inequality, and the last equality is derived from the linearity of expectation. Now, we compute the most inner term as

$$\left(\sum_{i=1}^{d} a_i u_i u_j\right)^2 = u_j^2 \sum_{k,l}^{d} a_k a_l u_k u_l$$

By independency of $u_i$ and $u_j$, we have

$$\mathbb{E}\left[\left(\sum_{i=1}^{d} a_i u_i u_j\right)^2\right] = \mathbb{E}\left[u_j^2 \sum_{k,l}^{d} a_k a_l u_k u_l\right]$$

$$= \mathbb{E}[a_j^2 u_j^4]$$

$$= 3a_j^2$$

since $\mathbb{E}[u_j^2 u_k u_l] = 0$ if either $k$ or $l$ is not equal to $j$ and $\mathbb{E}[u_j^4] = 3$. Thus, we have

$$\mathbb{E}\left[\|\langle\boldsymbol{a}, \boldsymbol{u}\rangle\boldsymbol{u}\|\right] = \sqrt{\sum_{j=1}^{d}\mathbb{E}\left[\left(\sum_{i=1}^{d} a_i u_i u_j\right)^2\right]}$$

$$\leq \sqrt{\sum_{j=1}^{d} 3a_j^2}$$

$$= \sqrt{3}\|\boldsymbol{a}\|$$

Similarly, we also have

$$\mathbb{E}\left[\|\langle\boldsymbol{a}, \boldsymbol{u}\rangle\boldsymbol{u}\|^2\right] = 3\|\boldsymbol{a}\|^2$$

$\square$

**Lemma C.7.** *The expectation of $\|\widehat{\boldsymbol{g}}_t\|$ is bounded by*

$$\mathbb{E}\big[\|\widehat{\boldsymbol{g}}_t\|\big]^2 \le \mathbb{E}\big[\|\widehat{\boldsymbol{g}}_t\|^2\big] \le \frac{\sqrt{105}\mu^2 L^2 r^2(d+4)}{2} + 6\|\nabla\mathcal{L}(\boldsymbol{\theta}_t;\mathcal{B}_t)\|^2$$

*Proof.* By definition of $\widehat{\boldsymbol{g}}_t$, we have

$$\|\widehat{\boldsymbol{g}}_t\|^2 = \left\|\frac{\mathcal{L}(\boldsymbol{\theta}_t+\mu\boldsymbol{u}_t;\mathcal{B}_t)-\mathcal{L}(\boldsymbol{\theta}_t-\mu\boldsymbol{u};\mathcal{B})}{2\mu}\boldsymbol{u}_t\right\|^2$$

$$\le \frac{2}{4\mu^2}\left\|\Big(\mathcal{L}(\boldsymbol{\theta}_t+\mu\boldsymbol{u}_t)-\mathcal{L}(\boldsymbol{\theta}_t)-\mu\langle\nabla\mathcal{L}(\boldsymbol{\theta}_t),\boldsymbol{u}_t\rangle\Big)\boldsymbol{u}_t - \Big(\mathcal{L}(\boldsymbol{\theta}_t-\mu\boldsymbol{u}_t)-\mathcal{L}(\boldsymbol{\theta}_t)-\mu\langle\nabla\mathcal{L}(\boldsymbol{\theta}_t),-\boldsymbol{u}_t\rangle\Big)\boldsymbol{u}_t\right\|^2$$

$$+ \frac{2}{4\mu^2}\|2\mu\langle\nabla\mathcal{L}(\boldsymbol{\theta}_t),\boldsymbol{u}_t\rangle\boldsymbol{u}_t\|^2$$

$$\le \frac{1}{\mu^2}\left\|\Big(\mathcal{L}(\boldsymbol{\theta}_t+\mu\boldsymbol{u}_t)-\mathcal{L}(\boldsymbol{\theta}_t)-\mu\langle\nabla\mathcal{L}(\boldsymbol{\theta}_t),\boldsymbol{u}_t\rangle\Big)\boldsymbol{u}_t\right\|^2 + \frac{1}{\mu^2}\left\|\Big(\mathcal{L}(\boldsymbol{\theta}_t-\mu\boldsymbol{u}_t)-\mathcal{L}(\boldsymbol{\theta}_t)-\mu\langle\nabla\mathcal{L}(\boldsymbol{\theta}_t),-\boldsymbol{u}_t\rangle\Big)\boldsymbol{u}_t\right\|^2$$

$$+ 2\|\langle\nabla\mathcal{L}(\boldsymbol{\theta}_t),\boldsymbol{u}_t\rangle\boldsymbol{u}_t\|^2$$

$$\le \frac{\mu^2 L^2}{2}\|\boldsymbol{u}_t\|^6 + 2\|\langle\nabla\mathcal{L}(\boldsymbol{\theta}_t),\boldsymbol{u}_t\rangle\boldsymbol{u}_t\|^2$$

The first and the second inequalities come from Young's inequality, and we use the smoothness condition in the last inequality as below. Thus, the expectation is

$$\mathbb{E}_{\boldsymbol{u}_t}[\|\widehat{\boldsymbol{g}}_t\|^2] \le \mathbb{E}\left[\frac{\mu^2 L^2}{2}\|\boldsymbol{u}_t\|^6 + 2\|\langle\nabla\mathcal{L}(\boldsymbol{\theta}_t),\boldsymbol{u}_t\rangle\boldsymbol{u}_t\|^2\right]$$

$$\le \frac{\mu^2 L^2(d+6)^3}{2} + 6\|\nabla\mathcal{L}(\boldsymbol{\theta}_t;\mathcal{B}_t)\|^2$$

where we use Lemma C.4 and Lemma C.6. Therefore, if we choose $\mu \le \frac{1}{(d+6)^{3/2}}$, the expected gradient norm is dimension-free! Also, if we assume bounded gradient, then we have

$$\mathbb{E}_{\boldsymbol{u},\boldsymbol{z}}[\|\widehat{\boldsymbol{g}}_t\|^2] \le \frac{\mu^2 L^2(d+6)^3}{2} + 6\mathbb{E}_{\boldsymbol{z}}[\|\nabla\mathcal{L}(\boldsymbol{\theta}_t;\mathcal{B}_t)\|^2]$$

$$\le \frac{\mu^2 L^2(d+6)^3}{2} + 6\left(G^2 + \frac{\sigma^2}{B}\right)$$

$$=: \widetilde{\sigma}(B)^2$$

$\square$

From the preceding lemma, we define an important quantity $\widetilde{\sigma}(B)$ for batch size $B$ as

$$\widetilde{\sigma}(B)^2 = \frac{\mu^2 L^2(d+6)^3}{2} + 6\left(G^2 + \frac{\sigma^2}{B}\right) \tag{1}$$

where $\sigma$ comes from standard bounded variance condition of first-order gradient.

**Lemma C.8.** *Under the bounded variance/gradient condition, i.e., $\mathbb{E}_{\boldsymbol{z}}[\|\nabla\mathcal{L}(\boldsymbol{\theta}_t;\boldsymbol{z})-\nabla\mathcal{L}(\boldsymbol{\theta}_t)\|^2] \le \sigma^2$ and $\|\nabla\mathcal{L}(\boldsymbol{\theta}_t)\| \le G$, we have for batch size $B$*

$$\mathbb{E}_{\boldsymbol{u},\boldsymbol{z}}[\|\widehat{\boldsymbol{m}}_t\|^2] \le \widetilde{\sigma}(B)^2$$

*Proof.* By mathematical induction and Jensen's inequality for convex function $\|\cdot\|^2$, it is easy to show that the inequality holds.

$$\mathbb{E}_{\boldsymbol{u},\boldsymbol{z}}[\|\widehat{\boldsymbol{m}}_t\|^2] = \mathbb{E}_{\boldsymbol{u},\boldsymbol{z}}[\|c_t\widehat{\boldsymbol{m}}_{t-1} + (1-c_t)\widehat{\boldsymbol{g}}_t\|^2]$$

$$\le c_t\mathbb{E}[\|\widehat{\boldsymbol{m}}_{t-1}\|^2] + (1-c_t)\mathbb{E}[\|\widehat{\boldsymbol{g}}_t\|^2]$$

$$\le c_t\widetilde{\sigma}(B)^2 + (1-c_t)\widetilde{\sigma}(B)^2$$

$$= \widetilde{\sigma}(B)^2$$

The first term is bounded by induction and the second term is bounded by the preceding lemma. $\square$

**Lemma C.9.** *Under the stepsize $\alpha \le \frac{(1-c_t)\delta}{L}$, the following inequalities hold*

$$\|\nabla\mathcal{L}_{\boldsymbol{u}}(\boldsymbol{\theta}_t)-\nabla\mathcal{L}_{\boldsymbol{u}}(\boldsymbol{\theta}_{t-1})\| \le (1-c_t)\widetilde{\sigma}(B)$$

$$\mathbb{E}[\|\widehat{\boldsymbol{g}}_t-\nabla\mathcal{L}_{\boldsymbol{u}}(\boldsymbol{\theta}_t)\|] \le \widetilde{\sigma}(B)$$

*Proof.* Under the stepsize $\alpha \leq \frac{(1-c_t)\delta}{L}$, the following inequalities hold

$$\|\nabla\mathcal{L}_{\boldsymbol{u}}(\boldsymbol{\theta}_t) - \nabla\mathcal{L}_{\boldsymbol{u}}(\boldsymbol{\theta}_{t-1})\|^2 \leq L^2\|\boldsymbol{\theta}_t - \boldsymbol{\theta}_{t-1}\|^2$$

$$\leq \frac{\alpha^2 L^2}{\delta^2}\|\widehat{\boldsymbol{m}}_t\|^2$$

Taking the expectation, we have

$$\|\nabla\mathcal{L}_{\boldsymbol{u}}(\boldsymbol{\theta}_t) - \nabla\mathcal{L}_{\boldsymbol{u}}(\boldsymbol{\theta}_{t-1})\|^2 \leq (1-c_t)^2\widetilde{\sigma}(B)^2$$

The ZO gradient variance is bounded by

$$\mathbb{E}[\|\widehat{\boldsymbol{g}}_t - \nabla\mathcal{L}_{\boldsymbol{u}}(\boldsymbol{\theta}_t)\|^2] = \mathrm{Var}(\widehat{\boldsymbol{g}}_t)$$

$$= \mathbb{E}[\|\widehat{\boldsymbol{g}}_t\|^2] - \underbrace{\|\mathbb{E}[\widehat{\boldsymbol{g}}_t]\|^2}_{\|\nabla\mathcal{L}_{\boldsymbol{u}}(\boldsymbol{\theta}_t)\|^2}$$

$$\leq \mathbb{E}[\|\widehat{\boldsymbol{g}}_t\|^2]$$

$$\leq \widetilde{\sigma}(B)^2$$

$\square$

**Lemma C.10.** *The quantities $\mathbb{E}[\|\boldsymbol{\epsilon}_t\|]$ and $\mathbb{E}[\|\boldsymbol{\xi}_t\|]$ are bounded by*

$$\mathbb{E}_{\boldsymbol{u},\boldsymbol{z}}[\|\boldsymbol{\epsilon}_t\|] \leq 2\widetilde{\sigma}(B)$$

$$\mathbb{E}_{\boldsymbol{u},\boldsymbol{z}}[\|\boldsymbol{\xi}_t\|] \leq 2\widetilde{\sigma}(B)$$

*Proof.* We use mathematical induction. By the definition of $\boldsymbol{\epsilon}_t$, we have

$$\mathbb{E}[\|\boldsymbol{\epsilon}_t\|] = \mathbb{E}[\|c_t(\boldsymbol{\epsilon}_{t-1} + \nabla\mathcal{L}_{\boldsymbol{u}}(\boldsymbol{\theta}_{t-1}) - \nabla\mathcal{L}_{\boldsymbol{u}}(\boldsymbol{\theta}_t)) + (1-c_t)(\widehat{\boldsymbol{g}}_t - \nabla\mathcal{L}_{\boldsymbol{u}}(\boldsymbol{\theta}_t))\|]$$

$$\leq c_t\mathbb{E}[\|\boldsymbol{\epsilon}_{t-1}\|] + c_t\mathbb{E}[\|\nabla\mathcal{L}_{\boldsymbol{u}}(\boldsymbol{\theta}_{t-1}) - \nabla\mathcal{L}(\boldsymbol{\theta}_t)\|] + (1-c_t)\mathbb{E}[\|\widehat{\boldsymbol{g}}_t - \nabla\mathcal{L}_{\boldsymbol{u}}(\boldsymbol{\theta}_t)\|]$$

$$\leq 2c_t\widetilde{\sigma}(B)^2 + c_t(1-c_t)\widetilde{\sigma}(B) + (1-c_t)\widetilde{\sigma}(B)$$

$$= (-c_t^2 + 2c_t + 1)\widetilde{\sigma}(B)$$

$$= 2\widetilde{\sigma}(B)$$

In the second inequality, we use the induction on $t-1$ and Lemma 25. Since $\|\boldsymbol{\xi}_{t-1}\| \leq \|\boldsymbol{\epsilon}_t\|$ and $\mathbb{E}[\|\boldsymbol{\epsilon}_t\|] \leq 2\widetilde{\sigma}(B)$ holds for all $t$, the bound for $\mathbb{E}[\|\boldsymbol{\xi}_t\|]$ is also trivial. $\square$

**Lemma C.11.** *Adam satisfies*

$$\sum_{t=1}^{T}\left(2\|\nabla\mathcal{L}_{\boldsymbol{u}}(\boldsymbol{\theta}_t)\|^2 - 2\kappa_\delta\mathbb{E}[\|\boldsymbol{\epsilon}_t\|^2]\right) \leq \frac{4(\sqrt{\zeta}+\delta)\Delta}{\alpha} + \frac{\alpha\kappa_\delta}{\delta}\sum_{t=1}^{T}\mathbb{E}\left[\widehat{\boldsymbol{m}}_t^\top\boldsymbol{H}(\boldsymbol{\theta}_t)\widehat{\boldsymbol{m}}_t\right]$$

*Proof.* For ease of notation, we let $(\widehat{\boldsymbol{V}}_t^{1/2} + \delta\boldsymbol{I}_d)^{-1} = \boldsymbol{\Lambda}_t$ and $\boldsymbol{\epsilon}_t = \widehat{\boldsymbol{m}}_t - \nabla\mathcal{L}_{\boldsymbol{u}}(\boldsymbol{\theta}_t)$. By the revised local effective rank condition, we have

$$\mathcal{L}_{\boldsymbol{u}}(\boldsymbol{\theta}_{t+1}) - \mathcal{L}_{\boldsymbol{u}}(\boldsymbol{\theta}_t) \leq \langle\nabla\mathcal{L}_{\boldsymbol{u}}(\boldsymbol{\theta}_t), \boldsymbol{\theta}_{t+1} - \boldsymbol{\theta}_t\rangle + \frac{\alpha^2}{2}\|\boldsymbol{\theta}_{t+1} - \boldsymbol{\theta}_t\|_{\boldsymbol{H}(\boldsymbol{\theta}_t)}^2$$

$$= -\alpha\nabla\mathcal{L}_{\boldsymbol{u}}(\boldsymbol{\theta}_t)^\top\boldsymbol{\Lambda}_t\widehat{\boldsymbol{m}}_t + \frac{\alpha^2}{2}\|\widehat{\boldsymbol{m}}_t\|_{\boldsymbol{\Lambda}_t\boldsymbol{H}(\boldsymbol{\theta}_t)\boldsymbol{\Lambda}_t}^2$$

$$\leq -\alpha\|\nabla\mathcal{L}_{\boldsymbol{u}}(\boldsymbol{\theta}_t)\|_{\boldsymbol{\Lambda}_t}^2 - \alpha\nabla\mathcal{L}_{\boldsymbol{u}}(\boldsymbol{\theta}_t)^\top\boldsymbol{\Lambda}_t\boldsymbol{\epsilon}_t + \frac{\alpha^2}{2}\|\widehat{\boldsymbol{m}}_t\|_{\boldsymbol{\Lambda}_t\boldsymbol{H}(\boldsymbol{\theta}_t)\boldsymbol{\Lambda}_t}^2$$

$$\leq -\alpha\|\nabla\mathcal{L}_{\boldsymbol{u}}(\boldsymbol{\theta}_t)\|_{\boldsymbol{\Lambda}_t}^2 + \alpha\Big(\frac{1}{2}\|\nabla\mathcal{L}_{\boldsymbol{u}}(\boldsymbol{\theta}_t)\|^2 + \frac{1}{2}\|\boldsymbol{\epsilon}_t\|^2\Big) + \frac{\alpha^2}{2}\|\widehat{\boldsymbol{m}}_t\|_{\boldsymbol{\Lambda}_t\boldsymbol{H}(\boldsymbol{\theta}_t)\boldsymbol{\Lambda}_t}^2$$

$$\leq -\frac{\alpha}{2}\|\nabla\mathcal{L}_{\boldsymbol{u}}(\boldsymbol{\theta}_t)\|_{\boldsymbol{\Lambda}_t}^2 + \frac{\alpha}{2}\|\boldsymbol{\epsilon}_t\|_{\boldsymbol{\Lambda}_t}^2 + \frac{\alpha^2}{2(\sqrt{\gamma}+\delta)^2}\|\widehat{\boldsymbol{m}}_t\|_{\boldsymbol{H}(\boldsymbol{\theta}_t)}^2$$

$$\leq -\frac{\alpha}{2(\sqrt{\zeta}+\delta)}\|\nabla\mathcal{L}_{\boldsymbol{u}}(\boldsymbol{\theta}_t)\|^2 + \frac{\alpha}{2(\sqrt{\gamma}+\delta)}\|\boldsymbol{\epsilon}_t\|^2 + \frac{\alpha^2}{2(\sqrt{\gamma}+\delta)^2}\widehat{\boldsymbol{m}}_t^\top\boldsymbol{H}(\boldsymbol{\theta}_t)\widehat{\boldsymbol{m}}_t$$

By telescoping the above inequality and taking expectation, we have

$$\sum_{t=1}^{T}\Big(\frac{\alpha}{2(\sqrt{\zeta}+\delta)}\|\nabla\mathcal{L}_{\boldsymbol{u}}(\boldsymbol{\theta}_t)\|^2 - \frac{\alpha}{2(\sqrt{\gamma}+\delta)}\mathbb{E}[\|\boldsymbol{\epsilon}_t\|^2]\Big) \leq \mathcal{L}_{\boldsymbol{u}}(\boldsymbol{\theta}_1) - \mathcal{L}_{\boldsymbol{u}}(\boldsymbol{\theta}_{T+1})$$

$$+ \frac{\alpha^2}{2(\sqrt{\gamma}+\delta)^2}\sum_{t=1}^{T}\mathbb{E}\left[\widehat{\boldsymbol{m}}_t^\top\boldsymbol{H}(\boldsymbol{\theta}_t)\widehat{\boldsymbol{m}}_t\right]$$

where we define $\Delta := \mathcal{L}_{\boldsymbol{u}}(\boldsymbol{\theta}_1) - \mathcal{L}_{\boldsymbol{u}}(\boldsymbol{\theta}^*)$. Hence, the above inequality could be rewritten as

$$\sum_{t=1}^{T}\left(2\|\nabla\mathcal{L}_{\boldsymbol{u}}(\boldsymbol{\theta}_t)\|^2 - 2\kappa_\delta\mathbb{E}\big[\|\boldsymbol{\epsilon}_t\|^2\big]\right) \le \frac{4(\sqrt{\zeta}+\delta)\Delta}{\alpha} + \frac{\alpha\kappa_\delta}{\delta}\sum_{t=1}^{T}\mathbb{E}\Big[\widehat{\boldsymbol{m}}_t^{\mathsf{T}}\boldsymbol{H}(\boldsymbol{\theta}_t)\widehat{\boldsymbol{m}}_t\Big]$$

$\square$

**Lemma C.12.**

$$\sum_{t=1}^{T}\mathbb{E}[\widehat{\boldsymbol{m}}_t^{\mathsf{T}}\boldsymbol{H}(\boldsymbol{\theta}_t)\widehat{\boldsymbol{m}}_t] \le 2\mu^2 L^3 r(d+8)^2 T + \frac{10Lr\sigma^2}{B}T + 10Lr\sum_{t=1}^{T}\|\nabla\mathcal{L}(\boldsymbol{\theta}_t)\|^2$$

*Proof.* By the definition of $\widehat{\boldsymbol{m}}_t$, we have

$$\begin{aligned}
\widehat{\boldsymbol{m}}_t &= \beta_1\widehat{\boldsymbol{m}}_{t-1} + (1-\beta_1)\widehat{\boldsymbol{g}}_t \\
&= \beta_1^2\widehat{\boldsymbol{m}}_{t-2} + \beta_1(1-\beta_1)\widehat{\boldsymbol{g}}_{t-1} + (1-\beta_1)\widehat{\boldsymbol{g}}_t \\
&= \beta_1^t\widehat{\boldsymbol{m}}_0 + \cdots + (1-\beta_1)\widehat{\boldsymbol{g}}_t \\
&= (1-\beta_1)\sum_{i=0}^{t}\beta_1^{t-i}\widehat{\boldsymbol{g}}_i
\end{aligned}$$

Hence, by Jensen's inequality, we obtain

$$\begin{aligned}
\|\widehat{\boldsymbol{m}}_t\|_{\boldsymbol{H}(\boldsymbol{\theta}_t)}^2 &= Z^2(1-\beta_1)^2\left\|\sum_{i=0}^{t}\frac{\beta_1^{t-i}}{Z}\widehat{\boldsymbol{g}}_i\right\|_{\boldsymbol{H}(\boldsymbol{\theta}_t)}^2 \\
&\le Z(1-\beta_1)^2\sum_{i=0}^{t}\frac{\beta_1^{t-i}}{Z}\|\widehat{\boldsymbol{g}}_i\|_{\boldsymbol{H}(\boldsymbol{\theta}_t)}^2 \\
&\le (1-\beta_1)\sum_{i=0}^{t}\beta_1^{t-i}\|\widehat{\boldsymbol{g}}_i\|_{\boldsymbol{H}(\boldsymbol{\theta}_t)}^2
\end{aligned}$$

where $Z = \sum_{i=0}^{t}\beta_1^{t-i} = \frac{1-\beta_1^t}{1-\beta_1} \le \frac{1}{1-\beta_1}$. Taking the expectation yields

$$\begin{aligned}
\mathbb{E}\left[\widehat{\boldsymbol{m}}_t^{\mathsf{T}}\boldsymbol{H}(\boldsymbol{\theta}_t)\widehat{\boldsymbol{m}}_t\right] &\le (1-\beta_1)\sum_{s=0}^{t}(\beta_1)^{t-s}\mathbb{E}\left[\widehat{\boldsymbol{g}}_s^{\mathsf{T}}\boldsymbol{H}(\boldsymbol{\theta}_t)\widehat{\boldsymbol{g}}_s\right] \\
&\le (1-\beta_1)\sum_{s=0}^{t}(\beta_1)^{t-s}\left(2\mu^2 L^3 r(d+8)^2 + 10Lr\|\nabla\mathcal{L}(\boldsymbol{\theta}_s;\mathcal{B}_s)\|^2\right) \\
&\le 2(1-\beta_1^{t+1})\mu^2 L^3 r(d+8)^2 + 10Lr(1-\beta_1)\sum_{s=0}^{t}(\beta_1^2)^{t-s}\|\nabla\mathcal{L}(\boldsymbol{\theta}_s;\mathcal{B}_s)\|^2 \\
&\le 2\mu^2 L^3 r(d+8)^2 + 10Lr(1-\beta_1)\sum_{s=0}^{t}(\beta_1)^{t-s}\left(\|\nabla\mathcal{L}(\boldsymbol{\theta}_s)\|^2 + \frac{\sigma^2}{B}\right)
\end{aligned}$$

where we use Lemma C.5. We compute the summation as

$$\sum_{s=0}^{t}(\beta_1^2)^{t-s}\|\nabla\mathcal{L}(\boldsymbol{\theta}_s)\|^2 = (\beta_1)^{t-1}\left(\|\nabla\mathcal{L}(\boldsymbol{\theta}_1)\|^2 + \frac{\sigma^2}{B}\right) + \cdots + \left(\|\nabla\mathcal{L}(\boldsymbol{\theta}_t)\|^2 + \frac{\sigma^2}{B}\right)$$

$$\sum_{s=0}^{t-1}(\beta_1)^{t-1-s}\|\nabla\mathcal{L}(\boldsymbol{\theta}_s)\|^2 = (\beta_1)^{t-2}\left(\|\nabla\mathcal{L}(\boldsymbol{\theta}_1)\|^2 + \frac{\sigma^2}{B}\right) + \cdots + \left(\|\nabla\mathcal{L}(\boldsymbol{\theta}_{t-1})\|^2 + \frac{\sigma^2}{B}\right)$$

$$\vdots$$

$$\sum_{s=0}^{1}(\beta_1)^{1-s}\|\nabla\mathcal{L}(\boldsymbol{\theta}_s)\|^2 = \|\nabla\mathcal{L}(\boldsymbol{\theta}_1)\|^2 + \frac{\sigma^2}{B}$$

Finally, we have

$$\sum_{t=0}^{T} \mathbb{E}\left[\widehat{\boldsymbol{m}}_t^{\mathsf{T}} \boldsymbol{H}(\boldsymbol{\theta}_t)\widehat{\boldsymbol{m}}_t\right] \leq 2\mu^2 L^3 r(d+8)^2 T + 10Lr(1-\beta_1)\sum_{t=0}^{T}\sum_{s=0}^{t}(\beta_1)^{t-s}\left(\|\nabla\mathcal{L}(\boldsymbol{\theta}_s)\|^2 + \frac{\sigma^2}{B}\right)$$

$$= 2\mu^2 L^3 r(d+8)^2 T + 10Lr(1-\beta_1)\sum_{s=1}^{T}\left(\|\nabla\mathcal{L}(\boldsymbol{\theta}_s)\|^2 + \frac{\sigma^2}{B}\right)\sum_{t=0}^{T-s}(\beta_1)^t$$

$$= 2\mu^2 L^3 r(d+8)^2 T + 10Lr\sum_{t=1}^{T}\left(\|\nabla\mathcal{L}(\boldsymbol{\theta}_t)\|^2 + \frac{\sigma^2}{B}\right)$$

$$\leq 2\mu^2 L^3 r(d+8)^2 T + \frac{10Lr\sigma^2}{B}T + 10Lr\sum_{t=1}^{T}\|\nabla\mathcal{L}(\boldsymbol{\theta}_t)\|^2$$

$$\square$$

**Lemma C.13.** *For $\boldsymbol{\epsilon}_t$, we have the following recursive relation*

$$\sum_{t=1}^{T-1}\left(\mathbb{E}[\|\boldsymbol{\epsilon}_t\|^2] - \frac{1}{2\kappa_\delta}\|\nabla\mathcal{L}_{\boldsymbol{u}}(\boldsymbol{\theta}_t)\|^2\right) \leq \frac{2\widetilde{\sigma}(B)^2}{1-\beta_1}(5 + 3(1-\beta_1)^2 T)$$

*Proof.* By the definition of $\boldsymbol{\epsilon}_t$, we have

$$\boldsymbol{\epsilon}_t = \boldsymbol{\xi}_{t-1} + (1-c_t)(\widehat{\boldsymbol{g}}_t - \nabla\mathcal{L}_{\boldsymbol{u}}(\boldsymbol{\theta}_t))$$

Also, we have

$$\|\nabla\mathcal{L}_{\boldsymbol{u}}(\boldsymbol{\theta}_t) - \nabla\mathcal{L}_{\boldsymbol{u}}(\boldsymbol{\theta}_{t-1})\| \leq L\|\boldsymbol{\theta}_t - \boldsymbol{\theta}_{t-1}\|$$

$$\leq \frac{\alpha L}{\delta}\|\widehat{\boldsymbol{m}}_{t-1}\|$$

$$\leq \frac{\alpha L}{\delta}(\|\boldsymbol{\epsilon}_{t-1}\| + \|\nabla\mathcal{L}_{\boldsymbol{u}}(\boldsymbol{\theta}_{t-1})\|)$$

Now, we compute the quantity $\|\boldsymbol{\xi}_{t-1}\|^2$ as

$$\|\boldsymbol{\xi}_{t-1}\|^2 = \|c_t\boldsymbol{\epsilon}_{t-1} + c_t(\nabla\mathcal{L}_{\boldsymbol{u}}(\boldsymbol{\theta}_{t-1}) - \nabla\mathcal{L}_{\boldsymbol{u}}(\boldsymbol{\theta}_t))\|^2$$

$$\leq c_t^2(2-c_t)\|\boldsymbol{\epsilon}_{t-1}\|^2 + c_t^2\left(1 + \frac{1}{1-c_t}\right)\|\nabla\mathcal{L}_{\boldsymbol{u}}(\boldsymbol{\theta}_t) - \nabla\mathcal{L}_{\boldsymbol{u}}(\boldsymbol{\theta}_{t-1})\|^2$$

$$\leq c_t\|\boldsymbol{\epsilon}_{t-1}\|^2 + \frac{1}{1-c_t}\|\nabla\mathcal{L}_{\boldsymbol{u}}(\boldsymbol{\theta}_t) - \nabla\mathcal{L}_{\boldsymbol{u}}(\boldsymbol{\theta}_{t-1})\|^2$$

$$\leq c_t\|\boldsymbol{\epsilon}_{t-1}\|^2 + \frac{2\alpha^2 L^2}{(1-\beta_1)\delta^2}\left(\|\boldsymbol{\epsilon}_{t-1}\|^2 + \|\nabla\mathcal{L}_{\boldsymbol{u}}(\boldsymbol{\theta}_{t-1})\|^2\right)$$

$$\leq \frac{1+c_t}{2}\|\boldsymbol{\epsilon}_{t-1}\|^2 + \frac{(1-\beta_1)}{4\kappa_\delta}\|\nabla\mathcal{L}_{\boldsymbol{u}}(\boldsymbol{\theta}_{t-1})\|^2$$

under the stepsize condition $\alpha \leq \frac{(1-\beta_1)\delta}{2\sqrt{2}\sqrt{\kappa_\delta}L}$. Hence, by the definition of $\boldsymbol{\epsilon}_t$, we have

$$\|\boldsymbol{\epsilon}_t\|^2 = \|\boldsymbol{\xi}_{t-1}\|^2 + 2(1-c_t)\langle\boldsymbol{\xi}_{t-1}, \widehat{\boldsymbol{g}}_t - \nabla\mathcal{L}_{\boldsymbol{u}}(\boldsymbol{\theta}_t)\rangle + (1-c_t)^2\|\widehat{\boldsymbol{g}}_t - \nabla\mathcal{L}_{\boldsymbol{u}}(\boldsymbol{\theta}_t)\|^2$$

Taking the expectation, we have

$$\mathbb{E}_{\boldsymbol{u},\boldsymbol{z}}\left[\|\boldsymbol{\epsilon}_t\|^2\right] \leq \frac{1+c_t}{2}\mathbb{E}\left[\|\boldsymbol{\epsilon}_{t-1}\|^2\right] + \frac{(1-\beta_1)}{4\kappa_\delta}\|\nabla\mathcal{L}_{\boldsymbol{u}}(\boldsymbol{\theta}_{t-1})\|^2 + (1-c_t)^2\widetilde{\sigma}(B)^2$$

$$+ 2(1-c_t)\mathbb{E}\left[\langle\boldsymbol{\xi}_{t-1}, \widehat{\boldsymbol{g}}_t - \nabla\mathcal{L}_{\boldsymbol{u}}(\boldsymbol{\theta}_t)\rangle\right]$$

$$\leq \frac{1+c_t}{2}\mathbb{E}\left[\|\boldsymbol{\epsilon}_{t-1}\|^2\right] + \frac{(1-\beta_1)}{4\kappa_\delta}\|\nabla\mathcal{L}_{\boldsymbol{u}}(\boldsymbol{\theta}_{t-1})\|^2 + (1-c_t)^2\widetilde{\sigma}(B)^2$$

The expectation of inner product $\mathbb{E}_{t-1}\left[\langle\boldsymbol{\xi}_{t-1}, \widehat{\boldsymbol{g}}_t - \nabla\mathcal{L}_{\boldsymbol{u}}(\boldsymbol{\theta}_t)\rangle\right] = 0$, we have $\mathbb{E}_{\boldsymbol{u},\boldsymbol{z}}[\langle\boldsymbol{\xi}_{t-1}, \widehat{\boldsymbol{g}}_t - \nabla\mathcal{L}_{\boldsymbol{u}}(\boldsymbol{\theta}_t)\rangle] = 0$. Thus, we have

$$\mathbb{E}_{\boldsymbol{u},\boldsymbol{z}}\left[\|\boldsymbol{\epsilon}_t\|^2\right] \leq \frac{1+c_t}{2}\mathbb{E}\left[\|\boldsymbol{\epsilon}_{t-1}\|^2\right] + \frac{1-\beta_1}{4\kappa_\delta}\|\nabla\mathcal{L}_{\boldsymbol{u}}(\boldsymbol{\theta}_{t-1})\|^2 + (1-c_t)^2\widetilde{\sigma}(B)^2$$

$$= \left(1 - \frac{1-c_t}{2}\right)\mathbb{E}\left[\|\boldsymbol{\epsilon}_{t-1}\|^2\right] + \frac{1-\beta_1}{4\kappa_\delta}\|\nabla\mathcal{L}_{\boldsymbol{u}}(\boldsymbol{\theta}_{t-1})\|^2 + (1-c_t)^2\widetilde{\sigma}(B)^2$$

By this relation, we have

$$\frac{1-\beta_1}{2}\mathbb{E}[\|\boldsymbol{\epsilon}_{t-1}\|^2] \le \frac{1-c_t}{2}\mathbb{E}[\|\boldsymbol{\epsilon}_{t-1}\|^2]$$

$$\le \mathbb{E}[\|\boldsymbol{\epsilon}_{t-1}\|^2] - \mathbb{E}[\|\boldsymbol{\epsilon}_t\|^2] + \frac{1-\beta_1}{4\kappa_\delta}\mathbb{E}[\|\nabla\mathcal{L}_{\boldsymbol{u}}(\boldsymbol{\theta}_{t-1})\|^2] + (1-c_t)^2\widetilde{\sigma}(B)^2$$

Hence, we have

$$\sum_{t=2}^{T+1}\left(\frac{1-\beta_1}{2}\mathbb{E}[\|\boldsymbol{\epsilon}_{t-1}\|^2] - \frac{1-\beta_1}{4\kappa_\delta}\mathbb{E}[\|\nabla\mathcal{L}_{\boldsymbol{u}}(\boldsymbol{\theta}_{t-1})\|^2]\right) \le \mathbb{E}[\|\boldsymbol{\epsilon}_1\|^2] - \mathbb{E}[\|\boldsymbol{\epsilon}_{T+1}\|^2] + \widetilde{\sigma}(B)^2\sum_{t=2}^{T+1}(1-c_t)^2$$

$$\le \mathbb{E}[\|\boldsymbol{\epsilon}_1\|^2] + \widetilde{\sigma}(B)^2\sum_{t=2}^{T+1}(1-c_t)^2$$

$$\le \widetilde{\sigma}(B)^2 + \widetilde{\sigma}(B)^2\Big(4 + 3(1-\beta_1)^2T\Big)$$

$$= 5\widetilde{\sigma}(B)^2 + 3(1-\beta_1)^2\widetilde{\sigma}(B)^2T$$

since $\boldsymbol{\epsilon}_1 = \widehat{\boldsymbol{g}}_1 - \nabla\mathcal{L}_{\boldsymbol{u}}(\boldsymbol{\theta}_1)$ and we use Lemma C.14. Therefore, we have

$$\sum_{t=1}^{T}\Big(2\kappa_\delta\mathbb{E}[\|\boldsymbol{\epsilon}_t\|] - \|\nabla\mathcal{L}_{\boldsymbol{u}}(\boldsymbol{\theta}_t)\|^2\Big) \le \frac{4\kappa_\delta\widetilde{\sigma}(B)^2}{1-\beta_1}(5 + 3(1-\beta_1)^2T)$$

$\square$

**Proof of Main Theorem.**  Now, we combine Lemma C.11 and Lemma C.13. Summing up the following inequalities.

$$\sum_{t=1}^{T}\Big(2\|\nabla\mathcal{L}_{\boldsymbol{u}}(\boldsymbol{\theta}_t)\|^2 - 2\kappa_\delta\mathbb{E}[\|\boldsymbol{\epsilon}_t\|^2]\Big) \le \frac{4(\sqrt{\zeta}+\delta)\Delta}{\alpha} + \frac{\alpha\kappa_\delta}{\delta}\sum_{t=1}^{T}\mathbb{E}\Big[\widehat{\boldsymbol{m}}_t^\mathsf{T}\boldsymbol{H}(\boldsymbol{\theta}_t)\widehat{\boldsymbol{m}}_t\Big]$$

$$\sum_{t=1}^{T}\Big(2\kappa_\delta\mathbb{E}[\|\boldsymbol{\epsilon}_t\|] - \|\nabla\mathcal{L}_{\boldsymbol{u}}(\boldsymbol{\theta}_t)\|^2\Big) \le \frac{4\kappa_\delta\widetilde{\sigma}(B)^2}{1-\beta_1}(5 + 3(1-\beta_1)^2T)$$

Then, we have

$$\sum_{t=1}^{T}\|\nabla\mathcal{L}_{\boldsymbol{u}}(\boldsymbol{\theta}_t)\|^2 \le \frac{4(\sqrt{\zeta}+\delta)\Delta}{\alpha} + \frac{\alpha\kappa_\delta}{\delta}\underbrace{\sum_{t=1}^{T}\mathbb{E}\Big[\widehat{\boldsymbol{m}}_t^\mathsf{T}\boldsymbol{H}(\boldsymbol{\theta}_t)\widehat{\boldsymbol{m}}_t\Big]}_{\text{Using Lemma C.12}} + \frac{4\kappa_\delta\widetilde{\sigma}(B)^2}{1-\beta_1}(5 + 3(1-\beta_1)^2T)$$

$$= \frac{4\Delta(\sqrt{\zeta}+\delta)}{\alpha} + \frac{20\kappa_\delta\widetilde{\sigma}(B)^2}{1-\beta_1} + 12\kappa_\delta\widetilde{\sigma}(B)^2(1-\beta_1)T$$

$$+ \frac{\alpha\kappa_\delta}{\delta}\Big(2\mu^2L^3r(d+8)^2T + \frac{10Lr(1-\beta_1)\sigma^2}{B}T + 10Lr(1-\beta_1)\sum_{t=1}^{T}\|\nabla\mathcal{L}(\boldsymbol{\theta}_t)\|^2\Big)$$

Using Lemma **??** and $\mu = \frac{1}{(d+8)\sqrt{T}}$, we obtain

$$\sum_{t=1}^{T}\|\nabla\mathcal{L}(\boldsymbol{\theta}_t)\|^2 \le \frac{3}{4}\mu^2L^2r^2dT + 2\sum_{t=1}^{T}\|\nabla\mathcal{L}_{\boldsymbol{u}}(\boldsymbol{\theta}_t)\|^2$$

$$\le \frac{3}{4}\mu^2L^2r^2dT + \frac{8\Delta(\sqrt{\zeta}+\delta)}{\alpha} + \frac{20\kappa_\delta\widetilde{\sigma}(B)^2}{1-\beta_1} + 12\kappa_\delta\widetilde{\sigma}(B)^2(1-\beta_1)T$$

$$+ \frac{2\alpha\kappa_\delta}{\delta}\Big(2\mu^2L^3r(d+8)^2T + \frac{10Lr(1-\beta_1)\sigma^2}{B}T + 10Lr(1-\beta_1)\sum_{t=1}^{T}\|\nabla\mathcal{L}(\boldsymbol{\theta}_t)\|^2\Big)$$

Again, we have

$$\left(1 - \frac{20\alpha\kappa_\delta Lr(1-\beta_1)}{\delta}\right)\sum_{t=1}^{T}\|\nabla\mathcal{L}(\boldsymbol{\theta}_t)\|^2 \le \frac{3}{4}\mu^2L^2r^2dT + \frac{8\Delta(\sqrt{\zeta}+\delta)}{\alpha} + \frac{20\kappa_\delta\widetilde{\sigma}(B)^2}{1-\beta_1} + 12\kappa_\delta\widetilde{\sigma}(B)^2(1-\beta_1)T$$

$$+ \frac{4\alpha\kappa_\delta\mu^2L^3r(d+8)^2}{\delta}T + \frac{20\alpha\kappa_\delta Lr(1-\beta_1)\sigma^2}{\delta B}T$$

According to Lemma C.9, Lemma C.13, and the above inequality, we should choose $\alpha$ as

$$\alpha \le \min\left\{ \frac{(1-c_t)\delta}{L}, \ \frac{(1-\beta_1)\delta}{2\sqrt{2}\sqrt{\kappa_\delta}L}, \ \frac{\delta}{20\kappa_\delta Lr(1-\beta_1)} \right\}$$

Note that we have

$$\frac{(1-\beta_1)\delta}{20\kappa_\delta Lr} \le \min\left\{ \frac{(1-c_t)\delta}{L}, \ \frac{(1-\beta_1)\delta}{2\sqrt{2}\sqrt{\kappa_\delta}L}, \ \frac{\delta}{20\kappa_\delta Lr(1-\beta_1)} \right\}$$

We choose the following parameter setting

$$\alpha \approx \frac{(1-\beta_1)\delta}{20\kappa_\delta Lr}, \quad 1-\beta_1 \le \min\left\{1, c_1\varepsilon^2\right\}, \quad T = \frac{1}{(1-\beta_1)^2} = \Omega(1/\varepsilon^4),$$

Under these parameters, we have

$$\left(1 - \frac{20\alpha\kappa_\delta Lr(1-\beta_1)}{\delta}\right) = 1 - (1-\beta_1)^2 \ge \beta_1$$

We also have by the condition on $\alpha$ and $T$

$$\frac{1}{\alpha T} \le \frac{\varepsilon^2}{c_2\delta}$$

for some constant $c_2$. Thus, the bound can be re-written as

$$\frac{\beta_1}{T}\sum_{t=1}^{T}\|\nabla\mathcal{L}(\boldsymbol{\theta}_t)\|^2 \le \frac{3}{4}\mu^2 L^2 r^2 d + \frac{8\Delta(\sqrt{\zeta}+\delta)}{\alpha T} + \frac{20\kappa_\delta\widetilde{\sigma}(B)^2}{(1-\beta_1)T} + 12\kappa_\delta\widetilde{\sigma}(B)^2(1-\beta_1)$$

$$+ \frac{4\alpha\kappa_\delta\mu^2 L^3 r(d+8)^2}{\delta} + \frac{20\alpha\kappa_\delta Lr(1-\beta_1)\sigma^2}{\delta B}$$

$$\le \frac{3}{4}\mu^2 L^2 r^2 d + \frac{8\Delta(\sqrt{\zeta}+\delta)}{c_2\delta}\varepsilon^2 + 32\kappa_\delta\widetilde{\sigma}(B)^2 c_1\varepsilon^2$$

$$+ \underbrace{\frac{4\alpha\kappa_\delta\mu^2 L^3 r(d+8)^2}{\delta}}_{Q} + \frac{20c_1\alpha\kappa_\delta Lr\sigma^2}{\delta B}\varepsilon^2$$

where $c_1$ and $c_2$ are constants independent of the problem dimension $d$. Here, $\widetilde{\sigma}(B)$ is defined as

$$\widetilde{\sigma}(B)^2 = \frac{\sqrt{105}\mu^2 L^2 r^2(d+4)}{2} + 6\left(G^2 + \frac{\sigma^2}{B}\right)$$

**Lemma C.14.** *For $c_t$ and $\beta_1$ defined in previous lemmas, we have*

$$\sum_{t=2}^{T+1}(1-c_t)^2 \le 4 + 3(1-\beta_1)^2 T$$

*Proof.* By the definition of $c_t$, we have

$$(1-c_t)^2 = \frac{(1-\beta_1)^2}{(1-\beta_1^t)^2}$$

Note that the following inequality holds for $x \ge 1$,

$$\left(1 - \frac{1}{x}\right)^x \le \frac{1}{e}.$$

Thus, we have for $x = \frac{1}{1-\beta_1}$

$$\beta_1^{\frac{1}{1-\beta_1}} \le \frac{1}{e}$$

Since $\beta_1 \le 1$, for any $t \ge \frac{1}{1-\beta_1}$, we have

$$\beta_1^t \le \beta_1^{\frac{1}{1-\beta_1}} \le \frac{1}{e}$$

For this range of $t$, we have

$$\sum_{\frac{1}{1-\beta_1} \leq t \leq T+1} (1-c_t)^2 = (1-\beta_1)^2 \sum_{\frac{1}{1-\beta_1} \leq t \leq T+1} \frac{1}{(1-\beta_1^t)^2}$$

$$\leq (1-\beta_1)^2 \sum_{t=2}^{T+1} \frac{1}{\left(1-\frac{1}{e}\right)^2}$$

$$\leq 3(1-\beta_1)^2 T$$

since $\frac{1}{\left(1-\frac{1}{e}\right)^2} \approx 2.50$. For $t < \frac{1}{1-\beta_1}$, we use the following two inequalities

$$(1-x)^r \leq e^{-rx}, \quad e^{-x} \leq 1 - x + \frac{x^2}{2}$$

Note that the first inequality holds for any $x \in \mathbb{R}$ with positive $r > 0$, and the second inequality holds for $x \geq 0$. Since $t < \frac{1}{1-\beta_1}$, $(1-\beta_1)t < 1$ holds, so again we have

$$\beta_1^t = (1 - (1-\beta_1))^t \leq e^{-(1-\beta_1)t} \leq 1 - (1-\beta_1)t + \frac{(1-\beta_1)^2 t^2}{2} \leq 1 - \frac{(1-\beta_1)t}{2}$$

In this range, we have

$$\sum_{2 \leq t < \frac{1}{1-\beta_1}} (1-c_t)^2 \leq (1-\beta_1)^2 \sum_{2 \leq t < \frac{1}{1-\beta_1}} \frac{1}{(1-\beta_1^t)^2}$$

$$\leq (1-\beta_1)^2 \sum_{t=2}^{T+1} \frac{4}{(1-\beta_1)^2 t^2}$$

$$= 4 \sum_{t=2}^{T+1} \frac{1}{t^2}$$

$$\leq 4 \int_1^{T+1} \frac{1}{x^2} \, dx$$

$$\leq 4$$

Combining two inequalities for each range, we have

$$\sum_{t=1}^{T} (1-c_t)^2 \leq 4 + 3(1-\beta_1)^2 T$$

$\square$

## D  PROOF OF PROPOSITION 4.1

We assume that $\delta \geq \sqrt{\gamma}$ since we would like to find the order of $\delta$ that makes $\kappa_\delta$ independent of the problem dimension $d$. Therefore, the parameter $\delta$ should be in order of at least $\mathcal{O}(\sqrt{\zeta})$. Hence, we need to find the bound for $\zeta$.

$$\zeta \leq \max_{t \in [T], i \in [d]} \widehat{\boldsymbol{g}}_{t,i}^2$$

The $\widehat{\boldsymbol{g}}_{t,i}^2$ could be bounded as

$$\widehat{\boldsymbol{g}}_{t,i}^2 = \left( \frac{\mathcal{L}(\boldsymbol{\theta}_t + \mu \boldsymbol{u}_t; \mathcal{B}_t) - \mathcal{L}(\boldsymbol{\theta}_t - \mu \boldsymbol{u}_t; \mathcal{B}_t)}{2\mu} \right)^2 \boldsymbol{u}_{t,i}^2$$

The coefficient could be bounded as

$$\left( \frac{\mathcal{L}(\boldsymbol{\theta}_t + \mu \boldsymbol{u}_t; \mathcal{B}_t) - \mathcal{L}(\boldsymbol{\theta}_t - \mu \boldsymbol{u}_t; \mathcal{B}_t)}{2\mu} \right)^2 \leq \frac{\mu L}{2} \|\boldsymbol{u}_t\|^2 + G \|\boldsymbol{u}_t\|$$

$$\leq \frac{\mu L d}{2} + G\sqrt{d}$$

*Proof.* Let $u_j$ be 1-dimensional standard Gaussian sample. Then, we have

$$\mathbb{P}\left[|u_j| \geq x\right] \leq 2 \exp\left(-\frac{x^2}{2}\right)$$

Thus, we have for $x = \sqrt{\frac{\xi}{d}}$

$$\mathbb{P}\left[|u_j| \geq \sqrt{\frac{\xi}{d}}\right] \leq 2 \exp\left(-\frac{\xi}{2d}\right)$$

By the bounded gradient assumption with differentiability of $\mathcal{L}$, we have

$$|\mathcal{L}_i(\boldsymbol{\theta}) - \mathcal{L}_i(\boldsymbol{\phi})| \leq G\|\boldsymbol{\theta} - \boldsymbol{\phi}\|$$

for any $i \in [n]$. Therefore, we have

$$|\widehat{\boldsymbol{g}}_{t,i}| \leq G\|\boldsymbol{u}_t\| u_{t,i}$$
$$\leq G\sqrt{d} u_{t,i}$$

Hence, we have

$$\mathbb{P}\left[|\widehat{\boldsymbol{g}}_{t,i}| \geq G\sqrt{\xi}\right] \leq \mathbb{P}\left[\sqrt{d}G|u_{t,i}| \geq G\sqrt{\xi}\right] \leq 2 \exp\left(-\frac{\xi}{2d}\right)$$

By plugging $\xi = 2d\log(2dT/\eta)$, we have

$$\mathbb{P}\left[|\widehat{\boldsymbol{g}}_{t,i}| \geq G\sqrt{2d\log\left(\frac{2dT}{\eta}\right)}\right] \leq \frac{\eta}{dT}$$

From this inequality, we have by union bound

$$\mathbb{P}\left[|\widehat{\boldsymbol{g}}_{t,i}| \geq G\sqrt{2d\log\left(\frac{2dT}{\eta}\right)}, \forall t \in [T], i \in [d]\right] = \mathbb{P}\left[|\widehat{\boldsymbol{g}}_{t,i}| \geq G\sqrt{2d\log\left(\frac{2dT}{\delta}\right)}, \forall t, i\right]$$

$$\leq \sum_{t \in [T]}\sum_{i \in [d]} \mathbb{P}\left[|\widehat{\boldsymbol{g}}_{t,i}| \geq G\sqrt{2d\log\left(\frac{2dT}{\eta}\right)}\right]$$

$$\leq dT \cdot \frac{\eta}{dT}$$

$$= \eta$$

Therefore, the parameter $\zeta$ is bounded by

$$\zeta \leq 2dG\log\left(\frac{2dT}{\eta}\right)$$

with probability at least $1 - \eta$ for given $\eta \in (0, 1)$. $\qquad\square$

## E    PROOF OF THEOREM 4.2 WITH NON-ZERO $\beta_1$ AND $\beta_2$

We revisit the parameter update rule of ZO-Adam as

$$g_t = \frac{\mathcal{L}(\theta_t + \mu u_t; z_{i_t}) - \mathcal{L}(\theta_t - \mu u_t; z_{i_t})}{2\mu} u_t$$

$$m_{t+1} = \beta_1 m_t + (1 - \beta_1)g_t$$

$$v_{t+1} = \beta_2 v_t + (1 - \beta_2)g_t^{\odot 2}$$

$$\theta_{t+1} = \theta_t - \alpha \frac{m_{t+1}}{\sqrt{v_{t+1} + \delta}}$$

Note that the zeroth-order gradient is computed on a single datapoint $z_{i_t}$. We define several key quantities for our analysis as below.

$$\mathcal{E}_{t_0} = \{\theta_t = \theta_{t_0}\} \text{ (the event)}$$

$$d_t = \|\theta_t - \theta'_t\|, \quad \Delta_t = \mathbb{E}\big[d_t | \mathcal{E}_{t_0}\big],$$

$$\varphi_t = \|m_t - m'_t\|, \quad \Phi_t = \mathbb{E}\big[\varphi_t | \mathcal{E}_{t_0}\big],$$

$$\sigma_t = \|v_t - v'_t\|, \quad \Sigma_t = \mathbb{E}\big[\sigma_t | \mathcal{E}_{t_0}\big].$$

By the condition (G-4), we have that the minimum/maximum eigenvalues of preconditioners are bounded as

$$\min_{t \in [T], i \in [d]} \{v_{t,i}, v'_{t,i}\} \geq \gamma$$

$$\max_{t \in [T], i \in [d]} \{v_{t,i}, v'_{t,i}\} \leq \zeta$$

The loss function $\mathcal{L}(\cdot, z)$ with respect to each data sample $z$ is assumed to be $G$-Lipschitz continuous and $L$-smooth.

### E.1    AUXILIARY LEMMAS FOR THEOREM 4.2

From now, we define some hyperparameters to be used in the following arguments.

1. The smoothing parameter $\mu = \frac{c_\mu}{(d + 16)^{3/2}} \lessapprox \frac{c_\mu}{d\sqrt{d}}$.

2. $1 - \beta_{1,t} = \frac{c_1}{nt}$

3. $1 - \beta_{2,t} = \frac{c_2}{dnt}$

4. $\delta = \delta_0 d^{2/3}$

for some positive constants $c_1$, $c_2$, and $c_\mu$. In this configuration, it can be seen that $\beta_2$ should be closer to 1 than $\beta_1$, which coincides with the practical case such as $(\beta_1, \beta_2) = (0.9, 0.999)$.

**Lemma E.1.** *For any $t > t_0$, the distance between the zeroth-order gradients, evaluated on the same datapoint $z_{i_t}$ but on the different parameters $\theta_t$ and $\theta'_t$, is bounded by*

$$\mathbb{E}\left[\left\|\widehat{\nabla}\mathcal{L}(\theta_t; z_{i_t}) - \widehat{\nabla}\mathcal{L}(\theta'_t; z_{i_t})\right\|\right] \leq \mu L(d + 3)^{3/2} + \sqrt{3}L\Delta_t$$

$$\leq c_\mu L + \sqrt{3}L\Delta_t$$

*Proof.* For simplicity, we denote $\mathcal{L}(\theta_t; z_{i_t}) =: \mathcal{L}_{i_t}(\theta_t)$. By the definition of ZO gradient, we have

$$\widehat{\nabla}\mathcal{L}_{i_t}(\theta_t) = \frac{\mathcal{L}_{i_t}(\theta_t + \mu u_t) - \mathcal{L}_{i_t}(\theta_t - \mu u_t)}{2\mu} u_t$$

$$= \frac{\mathcal{L}_{i_t}(\theta_t + \mu u_t) - \mathcal{L}_{i_t}(\theta_t) - \mu\langle\nabla\mathcal{L}_{i_t}(\theta_t), u_t\rangle}{2\mu} u_t$$

$$- \frac{\mathcal{L}_{i_t}(\theta_t - \mu u_t) - \mathcal{L}_{i_t}(\theta_t) - \mu\langle\nabla\mathcal{L}_{i_t}(\theta_t), -u_t\rangle}{2\mu} u_t$$

$$+ \langle\nabla\mathcal{L}_{i_t}(\theta_t), u_t\rangle u_t$$

Combined with the distance computed on the parameter $\theta_t'$, we obtain

$$\widehat{\nabla}\mathcal{L}_{i_t}(\theta_t) - \widehat{\nabla}\mathcal{L}_{i_t}(\theta_t') = \frac{\mathcal{L}_{i_t}(\theta_t + \mu u_t) - \mathcal{L}_{i_t}(\theta_t) - \mu\langle \nabla\mathcal{L}_{i_t}(\theta_t), u_t\rangle}{2\mu}u_t$$

$$- \frac{\mathcal{L}_{i_t}(\theta_t - \mu u_t) - \mathcal{L}_{i_t}(\theta_t) - \mu\langle \nabla\mathcal{L}_{i_t}(\theta_t), -u_t\rangle}{2\mu}u_t$$

$$- \frac{\mathcal{L}_{i_t}(\theta_t' + \mu u_t) - \mathcal{L}_{i_t}(\theta_t') - \mu\langle \nabla\mathcal{L}_{i_t}(\theta_t'), u_t\rangle}{2\mu}u_t$$

$$+ \frac{\mathcal{L}_{i_t}(\theta_t' - \mu u_t) - \mathcal{L}_{i_t}(\theta_t') - \mu\langle \nabla\mathcal{L}_{i_t}(\theta_t'), -u_t\rangle}{2\mu}u_t$$

$$+ \left\langle \nabla\mathcal{L}_{i_t}(\theta_t) - \nabla\mathcal{L}_{i_t}(\theta_t'), u_t\right\rangle u_t$$

By the smoothness condition, the first four terms could be bounded by

$$\left\| \frac{\mathcal{L}_{i_t}(\theta_t + \mu u_t) - \mathcal{L}_{i_t}(\theta_t) - \mu\langle \nabla\mathcal{L}_{i_t}(\theta_t), u_t\rangle}{2\mu}u_t \right\| \le \frac{\mu L}{4}\|u_t\|^3$$

Hence, the distance we are interested in could be bounded by

$$\left\| \widehat{\nabla}\mathcal{L}_{i_t}(\theta_t) - \widehat{\nabla}\mathcal{L}_{i_t}(\theta_t') \right\| \le \mu L\|u_t\|^3 + \left\| \langle \nabla\mathcal{L}_{i_t}(\theta_t) - \nabla\mathcal{L}_{i_t}(\theta_t'), u_t\rangle u_t \right\|$$

Taking the expectation using Lemma C.3 yields that

$$\mathbb{E}_u\left[ \left\| \widehat{\nabla}\mathcal{L}_{i_t}(\theta_t) - \widehat{\nabla}\mathcal{L}_{i_t}(\theta_t') \right\| \right] \le \mu L \mathbb{E}_u\left[ \|u_t\|^3 \right] + \mathbb{E}_u\left[ \left\| \langle \nabla\mathcal{L}_{i_t}(\theta_t) - \nabla\mathcal{L}_{i_t}(\theta_t'), u_t\rangle u_t\right\| \right]$$

$$\le \mu L(d+3)^{3/2} + \sqrt{3}L\|\theta_t - \theta_t'\|$$

Therefore, we have for $\mu = \dfrac{c_\mu}{(d+16)^{3/2}}$

$$\mathbb{E}_u\left[ \left\| \widehat{\nabla}\mathcal{L}_{i_t}(\theta_t) - \widehat{\nabla}\mathcal{L}_{i_t}(\theta_t') \right\| \Big| \mathcal{E}_{t_0} \right] \le c_\mu L + \sqrt{3}L\Delta_t$$

$\square$

**Lemma E.2.** *The norm of ZO gradient is bounded by*

$$\mathbb{E}\left[ \left\| \widehat{\nabla}\mathcal{L}_{i_t}(\theta_t) \right\| \right] \le \frac{\mu L(d+3)^{3/2}}{2} + \sqrt{3}G$$

$$\le \frac{c_\mu L}{2} + \sqrt{3}G$$

$$\mathbb{E}\left[ \left\| \widehat{\nabla}\mathcal{L}_{i_t}(\theta_t) \right\|^2 \right] \le \frac{3\mu^2 L^2}{8}(d+6)^3 + 9G^2$$

$$\le \frac{3}{8}c_\mu^2 L^2 + 9G^2$$

*Proof.* By the definition of ZO gradient, we have

$$\left\| \widehat{\nabla}\mathcal{L}_{i_t}(\theta_t) \right\| \le \left\| \frac{\mathcal{L}_{i_t}(\theta_t + \mu u_t) - \mathcal{L}_{i_t}(\theta_t) - \mu\langle \nabla\mathcal{L}_{i_t}(\theta_t), u_t\rangle}{2\mu}u_t \right\|$$

$$+ \left\| \frac{\mathcal{L}_{i_t}(\theta_t - \mu u_t) - \mathcal{L}_{i_t}(\theta_t) - \mu\langle \nabla\mathcal{L}_{i_t}(\theta_t), -u_t\rangle}{2\mu}u_t \right\|$$

$$+ \left\| \langle \nabla\mathcal{L}_{i_t}(\theta_t), u_t\rangle u_t \right\|$$

From the above inequality, we have

$$\left\| \widehat{\nabla}\mathcal{L}_{i_t}(\theta_t) \right\|^2 \le 3\left\| \frac{\mathcal{L}_{i_t}(\theta_t + \mu u_t) - \mathcal{L}_{i_t}(\theta_t) - \mu\langle \nabla\mathcal{L}_{i_t}(\theta_t), u_t\rangle}{2\mu}u_t \right\|^2$$

$$+ 3\left\| \frac{\mathcal{L}_{i_t}(\theta_t - \mu u_t) - \mathcal{L}_{i_t}(\theta_t) - \mu\langle \nabla\mathcal{L}_{i_t}(\theta_t), -u_t\rangle}{2\mu}u_t \right\|^2$$

$$+ 3\left\| \langle \nabla\mathcal{L}_{i_t}(\theta_t), u_t\rangle u_t \right\|^2$$

$$\le \frac{3\mu^2 L^2}{8}\|u_t\|^6 + 3\|\langle \nabla\mathcal{L}_{i_t}(\theta_t), u_t\rangle u_t\|^2$$

Thus, we have the expectation

$$\mathbb{E}\left[\left\|\widehat{\nabla}\mathcal{L}_{i_t}(\theta_t)\right\|\right] \leq \frac{\mu L}{2}\mathbb{E}\left[\|u_t\|^3\right] + \mathbb{E}\left[\|\langle\nabla\mathcal{L}_{i_t}(\theta_t), u_t\rangle u_t\|\right]$$

$$\leq \frac{\mu L(d+3)^{3/2}}{2} + \sqrt{3}\|\nabla\mathcal{L}_{i_t}(\theta_t)\|$$

$$\leq \frac{\mu L(d+3)^{3/2}}{2} + \sqrt{3}G$$

$$= \frac{c_\mu L}{2} + \sqrt{3}G$$

and

$$\mathbb{E}\left[\left\|\widehat{\nabla}\mathcal{L}_{i_t}(\theta_t)\right\|^2\right] \leq \frac{3\mu^2 L^2}{8}(d+6)^3 + 9\|\nabla\mathcal{L}_{i_t}(\theta_t)\|^2$$

$$\leq \frac{3\mu^2 L^2}{8}(d+6)^3 + 9G^2$$

$$\leq \frac{3}{8}c_\mu^2 L^2 + 9G^2$$

Under our parameter settings of $\mu$, we finally get the results. $\qquad\square$

**Lemma E.3.** *The bound for $j$-th coordinate*

$$\mathbb{E}\left[\widehat{\nabla}\mathcal{L}_{i_t}(\theta_t)_j^4\right] \leq 81\left(\frac{1}{8}\mu^4 L^4(d+16)^4 + 4\sqrt{105}G^4\right)$$

$$\leq 81\left(\frac{L^4}{8n^4} + 4\sqrt{105}G^4\right)$$

$$\mathbb{E}\left[\widehat{\nabla}\mathcal{L}_{i_t}(\theta_t)_j^2\right] \leq 9\left(\frac{1}{2\sqrt{2}}\mu^2 L^2(d+16)^2 + 2\sqrt[4]{105}G^2\right)$$

$$\leq 9\left(\frac{L^2}{8\sqrt{2}} + 2\sqrt[4]{105}G^2\right)$$

*Proof.* Using the definition of ZO gradient esimate, we have

$$\mathbb{E}\left[\widehat{\nabla}\mathcal{L}_{i_t}(\theta_t)_j^4\right] = \mathbb{E}\left[\left(\frac{\mathcal{L}_{i_t}(\theta_t + \mu u_t) - \mathcal{L}_{i_t}(\theta_t - \mu u_t)}{2\mu}\right)^4 u_{t,j}^4\right]$$

$$\leq \sqrt{\mathbb{E}\left[\left(\frac{\mathcal{L}_{i_t}(\theta_t + \mu u_t) - \mathcal{L}_{i_t}(\theta_t - \mu u_t)}{2\mu}\right)^8\right]}\underbrace{\sqrt{\mathbb{E}\left[u_{t,j}^8\right]}}_{\leq 9^2}$$

The first mulplicant is bounded by

$$\mathbb{E}\left[(\mathcal{L}_{i_t}(\theta_t + \mu u_t) - \mathcal{L}_{i_t}(\theta_t - \mu u_t))^8\right] \leq \mathbb{E}\left[8\mu^{16}L^8\|u_t\|^{16} + 2048(\mu\langle\nabla\mathcal{L}_{i_t}(\boldsymbol{\theta}_t), u_t\rangle^8)\right]$$

$$\leq 8\mu^{16}L^8(d+16)^8 + 2048 \cdot 105\mu^8\|\nabla\mathcal{L}_{i_t}(\theta_t)\|^8$$

$$\leq 8\mu^{16}L^8(d+16)^8 + 2048 \cdot 105\mu^8 G^8$$

Hence, we have

$$\sqrt{\mathbb{E}\left[\left(\frac{\mathcal{L}_{i_t}(\theta_t + \mu u_t) - \mathcal{L}_{i_t}(\theta_t - \mu u_t)}{2\mu}\right)^8\right]} \leq \sqrt{\frac{1}{256\mu^8}(8\mu^{16}L^8(d+16)^8 + 2048 \cdot 105\mu^8 G^8)}$$

$$\leq \sqrt{\frac{1}{256\mu^8}8\mu^{16}L^8(d+16)^8} + \sqrt{\frac{1}{256\mu^8}2048 \cdot 105\mu^8 G^8}$$

$$= \frac{1}{8}\mu^4 L^4(d+16)^4 + 4\sqrt{105}G^4$$

Finally, the $j$-th coordinate of the zeroth-order gradient is bounded by

$$\mathbb{E}\left[\widehat{\nabla}\mathcal{L}_{i_t}(\theta_t)_j^4\right] \leq 81\left(\frac{1}{8}\mu^4 L^4(d+16)^4 + 4\sqrt{105}G^4\right)$$

$$\leq 81\left(\frac{c_\mu^4 L^4}{8(d+16)^2} + 4\sqrt{105}G^4\right)$$

Applying $\mu = \frac{c_\mu}{(d+16)^{3/2}}$, we have the final results. Lastly, we have

$$\mathbb{E}\left[\widehat{\nabla}\mathcal{L}_{i_t}(\theta_t)_j^2\right] \le 9\left(\frac{c_\mu^2 L^2}{2\sqrt{2}(d+16)} + 2\sqrt[4]{105}G^2\right)$$

$\square$

**Lemma E.4.** *Under the assumption $\|u_t\| \le \sqrt{d}$, the norms of ZO gradient and the momentum (without the expectation) are bounded by*

$$\|g_t\| \le dG$$
$$\|m_t\| \le dG$$

*Proof.* We use mathematical induction. For $t = 1$, the momentum is nothing but the scaled zeroth-order gradient as

$$\|m_1\| = \|\beta_1 m_0 + (1-\beta_1)\widehat{\nabla}\mathcal{L}_{i_1}(\theta_1)\|$$
$$\le (1-\beta_1)\|\widehat{\nabla}\mathcal{L}_{i_1}(\theta_1)\|$$

The size of zeroth-order gradient could be bounded by

$$\left\|\widehat{\nabla}\mathcal{L}_{i_1}(\theta_1)\right\| = \left\|\frac{\mathcal{L}_{i_1}(\theta_1 + \mu u_1) - \mathcal{L}_{i_1}(\theta_1 - \mu u_1)}{2\mu}u_1\right\|$$
$$\le G\left\|\frac{\theta_1 + \mu u_1 - (\theta_1 - \mu u_1)}{2\mu}u_1\right\|$$
$$= G\|u_1\|^2$$
$$\le dG$$

since we assume $G$-Lipschitz continuity. For the initial condition, we have

$$\|m_1\| = (1-\beta_1)\left\|\widehat{\nabla}\mathcal{L}_{i_1}(\theta_1)\right\|$$
$$\le dG$$

By the induction, we have

$$\|m_t\| \le \beta_1\|m_{t-1}\| + (1-\beta_1)\left\|\widehat{\nabla}\mathcal{L}_{i_t}(\theta_t)\right\|$$
$$\le \beta_1 dG + (1-\beta_1)dG$$
$$\le dG$$

$\square$

**Lemma E.5.** *The norm of preconditioner (vector) is bounded by*

$$\mathbb{E}\left[\|v_t\|\right] \le$$

*Proof.* We use induction. By the definition of $v_t$, we have

$$\|v_t\|^2 = \sum_{j=1}^d v_{t,j}^2$$
$$= \sum_{j=1}^d \left(\beta_2 v_{t-1,j} + (1-\beta_2)g_{t,j}^2\right)^2$$

For $t = 1$, we have

$$\|v_1\|^2 = \sum_{j=1}^d v_{1,j}^2$$
$$= (1-\beta_2)^2 \sum_{j=1}^d g_{1,j}^4$$

Hence, we have by Lemma E.3

$$\mathbb{E}\left[\|v_1\|^2\right] \leq (1-\beta_2)^2 \sum_{j=1}^{d} \mathbb{E}\left[g_{1,j}^4\right]$$

$$\leq 81d \left(\frac{c_\mu^4 L^4}{8(d+16)^2} + 4\sqrt{105}G^4\right)$$

For $v_t$, we have

$$\|v_t\|^2 = \sum_{j=1}^{d} v_{t,j}^2$$

$$= \sum_{j=1}^{d} \left(\beta_2 v_{t-1,j} + (1-\beta_2)g_{t,j}^2\right)^2$$

$$\leq \sum_{j=1}^{d} \left(\beta_2 v_{t-1,j}^2 + (1-\beta_2)g_{t,j}^4\right)$$

$$= \beta_2\|v_{t-1}\|^2 + (1-\beta_2)\sum_{j=1}^{d} g_{t,j}^4$$

Finally, by induction, we have

$$\mathbb{E}\left[\|v_t\|^2\right] \leq \beta_2\mathbb{E}\left[\|v_{t-1}\|^2\right] + (1-\beta_2)\sum_{j=1}^{d} \mathbb{E}\left[g_{t,j}^4\right]$$

$$\leq \beta_2 81d \left(\frac{c_\mu^4 L^4}{8(d+16)^2} + 4\sqrt{105}G^4\right) + (1-\beta_2)81d \left(\frac{c_\mu^4 L^4}{8(d+16)^2} + 4\sqrt{105}G^4\right)$$

$$\leq 81d \left(\frac{c_\mu^4 L^4}{8(d+16)^2} + 4\sqrt{105}G^4\right)$$

The expectation of norm of $v_t$ is bounded by

$$\mathbb{E}\left[\|v_t\|\right]^2 \leq \mathbb{E}\left[\|v_t\|^2\right]$$

$$\leq 81d \left(\frac{c_\mu^4 L^4}{8(d+16)^2} + 4\sqrt{105}G^4\right)$$

Hence, we obtain

$$\mathbb{E}\left[\|v_t\|\right] \leq 9\sqrt{d} \left(\frac{c_\mu^2 L^2}{2\sqrt{2}(d+16)} + 2\sqrt[4]{105}G^2\right)$$

$\square$

## E.2 Recursive Relations for $(\Delta_t, \Phi_t, \Sigma_t)$

Recall the parameter configurations as

$$\mu = \frac{c_\mu}{(d+16)^{3/2}}$$

$$\beta_{1,t} = 1 - \frac{\beta_1}{nt}$$

$$\beta_{2,t} = 1 - \frac{\beta_2}{dnt}$$

$$\delta = \delta_0 d^{2/3}$$

**Lemma E.6.** *The following recursive relation should hold that*

$$\Delta_{t+1} \leq \ ???$$

*Proof.* For any $t > t_0$, we have that

$$d_{t+1} = \left\| \theta_t - \alpha \frac{m_{t+1}}{\sqrt{v_{t+1}} + \delta} - \theta_t' + \alpha \frac{m_{t+1}'}{\sqrt{v_{t+1}'} + \delta} \right\|$$

$$\leq \|\theta_t - \theta_t'\| + \alpha \underbrace{\left\| \frac{m_{t+1}}{\sqrt{v_{t+1}} + \delta} - \frac{m_{t+1}'}{\sqrt{v_{t+1}'} + \delta} \right\|}_{R_t}$$

$$= \|\theta_t - \theta_t'\| + \alpha R_t$$

All the norm is $\ell_2$-norm. We consider two cases for recursive relation for $R_t$.

**Case 1:** With probability $1 - \frac{1}{n}$, we have $z_{i_t} = z_{i_t}'$ where the same datapoint is sampled at time $t$. Thus, we have

$$R_t = \left\| \frac{m_{t+1}}{\sqrt{v_{t+1}} + \delta} - \frac{m_{t+1}'}{\sqrt{v_{t+1}'} + \delta} \right\|$$

$$\leq \underbrace{\left\| \frac{m_{t+1}}{\sqrt{v_{t+1}} + \delta} - \frac{m_{t+1}'}{\sqrt{v_{t+1}} + \delta} \right\|}_{I_t} + \underbrace{\left\| \frac{m_{t+1}'}{\sqrt{v_{t+1}} + \delta} - \frac{m_{t+1}'}{\sqrt{v_{t+1}'} + \delta} \right\|}_{J_t}$$

We first bound the term $I_t$ using the condition for eigenvalues of $v_t$ and $v_t'$ by

$$I_t \leq \frac{1}{\sqrt{\gamma} + \delta} \|m_{t+1} - m_{t+1}'\|$$

Using Lemma E.1, the bound for $\|m_{t+1} - m_{t+1}'\|$ given $z_{i_t} = z_{i_t}'$ would be

$$\|m_{t+1} - m_{t+1}'\| = \left\| \beta_1 m_t + (1 - \beta_1)\widehat{\nabla}\mathcal{L}(\theta_t; z_{i_t}) - \beta_1 m_t' - (1 - \beta_1)\widehat{\nabla}\mathcal{L}(\theta_t'; z_{i_t}) \right\|$$

$$\leq \beta_1 \|m_t - m_t'\| + (1 - \beta_1) \left\| \widehat{\nabla}\mathcal{L}(\theta_t; z_{i_t}) - \widehat{\nabla}\mathcal{L}(\theta_t'; z_{i_t}) \right\|$$

Taking the expectation yields that

$$\mathbb{E}\left[ \|m_{t+1} - m_{t+1}'\|_2 | \{z_{i_t} = z_{i_t}'\} \right] \leq \beta_1 \Phi_t + (1 - \beta_1)\left( c_\mu L + \sqrt{3}L\Delta_t \right)$$

Hence, we have

$$\mathbb{E}\left[ I_t | \{z_{i_t} = z_{i_t}'\} \right] \leq \frac{\beta_1 \Phi_t + (1 - \beta_1)\left( c_\mu L + \sqrt{3}L\Delta_t \right)}{\sqrt{\gamma} + \delta}$$

$$\leq \frac{\beta_1 \Phi_t + (1 - \beta_1)\left( c_\mu L + \sqrt{3}L\Delta_t \right)}{\delta}$$

The bound for $J_t$ can be computed as

$$J_t = \left\| \frac{m_{t+1}'}{\sqrt{v_{t+1}} + \delta} - \frac{m_{t+1}'}{\sqrt{v_{t+1}'} + \delta} \right\|$$

$$= \left\| \frac{1}{\sqrt{v_{t+1}} + \delta} - \frac{1}{\sqrt{v_{t+1}'} + \delta} \right\| \|m_{t+1}'\|$$

$$\leq dG \left\| \frac{1}{\sqrt{v_{t+1}} + \delta} - \frac{1}{\sqrt{v_{t+1}'} + \delta} \right\|$$

where the last inequality comes from Lemma E.4. The quantity we are interested in is the expectation of $J_t$, which is given by

$$\mathbb{E}\left[ J_t \right] = \mathbb{E}\left[ \left\| \frac{1}{\sqrt{v_{t+1}} + \delta} - \frac{1}{\sqrt{v_{t+1}'} + \delta} \right\| \|m_{t+1}'\| \right]$$

$$\leq dG \mathbb{E}\left[ \left\| \frac{1}{\sqrt{v_{t+1}} + \delta} - \frac{1}{\sqrt{v_{t+1}'} + \delta} \right\| \right]$$

where we use Jensen's inequality for the expectation and Lemma E.4. The remaining term could be bounded as

$$
\frac{1}{\sqrt{v_{t+1,j} + \delta}} - \frac{1}{\sqrt{v'_{t+1,j} + \delta}} = \frac{\sqrt{v'_{t+1,j}} - \sqrt{v_{t+1,j}}}{\left(\sqrt{v_{t+1,j} + \delta}\right)\left(\sqrt{v'_{t+1,j} + \delta}\right)}
$$
$$
\leq \frac{1}{\delta}\left(\sqrt{v'_{t+1,j} + \delta} - \sqrt{v_{t+1,j} + \delta}\right)
$$
$$
= \frac{1}{\delta}\frac{\left(\sqrt{v'_{t+1,j} + \delta}\right)^2 - \left(\sqrt{v_{t+1,j} + \delta}\right)^2}{(\sqrt{v'_{t+1,j} + \delta}) + (\sqrt{v_{t+1,j} + \delta})}
$$
$$
\leq \frac{1}{2\delta^{3/2}}\left(v'_{t+1,j} - v_{t+1,j}\right)
$$

Hence, the following norm is bounded by

$$
\mathbb{E}\left[\left\|\frac{1}{\sqrt{v_{t+1} + \delta}} - \frac{1}{\sqrt{v'_{t+1} + \delta}}\right\|\right] = \mathbb{E}\left[\left(\sum_{j=1}^{d}\left(\frac{1}{\sqrt{v_{t+1,j} + \delta}} - \frac{1}{\sqrt{v'_{t+1,j} + \delta}}\right)^2\right)^{1/2}\right]
$$
$$
\leq \frac{1}{2\delta^{3/2}}\mathbb{E}\left[\|v_{t+1} - v'_{t+1}\|\right]
$$

Note that we have been considering the case of $z_{i_t} = z'_{i_t}$, the bound for the distance $\|v_{t+1} - v'_{t+1}\|$ in this case could be simplified as

$$
\|v_{t+1} - v'_{t+1}\| = \left\|\beta_2 v_t + (1 - \beta_2)\widehat{\nabla}\mathcal{L}_{i_t}(\theta_t)^2 - \beta_2 v'_t - (1 - \beta_2)\widehat{\nabla}\mathcal{L}_{i_t}(\theta'_t)^2\right\|
$$
$$
\leq \beta_2\|v_t - v'_t\| + (1 - \beta_2)\underbrace{\left\|\widehat{\nabla}\mathcal{L}_{i_t}(\theta_t)^2 - \widehat{\nabla}\mathcal{L}_{i_t}(\theta'_t)^2\right\|}_{N_t}
$$

The bound for $N_t$ is

$$
N_t = \left(\sum_{j=1}^{d}\left(\widehat{\nabla}\mathcal{L}_{i_t}(\theta_t)_j^2 - \widehat{\nabla}\mathcal{L}_{i_t}(\theta'_t)_j^2\right)^2\right)^{1/2}
$$
$$
= \left(\sum_{j=1}^{d}\left(\widehat{\nabla}\mathcal{L}_{i_t}(\theta_t)_j + \widehat{\nabla}\mathcal{L}_{i_t}(\theta'_t)_j\right)^2\left(\widehat{\nabla}\mathcal{L}_{i_t}(\theta_t)_j - \widehat{\nabla}\mathcal{L}_{i_t}(\theta'_t)_j\right)^2\right)^{1/2}
$$
$$
\leq \left(\sum_{j=1}^{d}\left(2\widehat{\nabla}\mathcal{L}_{i_t}(\theta_t)_j^2 + 2\widehat{\nabla}\mathcal{L}_{i_t}(\theta'_t)_j^2\right)\left(\widehat{\nabla}\mathcal{L}_{i_t}(\theta_t)_j - \widehat{\nabla}\mathcal{L}_{i_t}(\theta'_t)_j\right)^2\right)^{1/2}
$$

Therefore, the expectation of $N_t$ is

$$
\mathbb{E}\left[N_t\right] \leq \mathbb{E}\left[\left(\sum_{j=1}^{d}\left(2\widehat{\nabla}\mathcal{L}_{i_t}(\theta_t)_j^2 + 2\widehat{\nabla}\mathcal{L}_{i_t}(\theta'_t)_j^2\right)\left(\widehat{\nabla}\mathcal{L}_{i_t}(\theta_t)_j - \widehat{\nabla}\mathcal{L}_{i_t}(\theta'_t)_j\right)^2\right)^{1/2}\right]
$$
$$
\leq \mathbb{E}\left[\left(\sum_{j=1}^{d}\left(2\widehat{\nabla}\mathcal{L}_{i_t}(\theta_t)_j^2 + 2\widehat{\nabla}\mathcal{L}_{i_t}(\theta'_t)_j^2\right)^2\right)^{1/4}\left(\sum_{j=1}^{d}\left(\widehat{\nabla}\mathcal{L}_{i_t}(\theta_t)_j - \widehat{\nabla}\mathcal{L}_{i_t}(\theta'_t)_j\right)^4\right)^{1/4}\right]
$$
$$
\leq \mathbb{E}\left[\left(\sum_{j=1}^{d}\left(4\widehat{\nabla}\mathcal{L}_{i_t}(\theta_t)_j^4 + 4\widehat{\nabla}\mathcal{L}_{i_t}(\theta'_t)_j^4\right)\right)^{1/4}\left(\sum_{j=1}^{d}\left(\widehat{\nabla}\mathcal{L}_{i_t}(\theta_t)_j - \widehat{\nabla}\mathcal{L}_{i_t}(\theta'_t)_j\right)^2\right)^{1/2}\right]
$$
$$
\leq \mathbb{E}\left[\left(\sum_{j=1}^{d}\left(2\widehat{\nabla}\mathcal{L}_{i_t}(\theta_t)_j^2 + 2\widehat{\nabla}\mathcal{L}_{i_t}(\theta'_t)_j^2\right)\right)^{1/2}\left(\sum_{j=1}^{d}\left(\widehat{\nabla}\mathcal{L}_{i_t}(\theta_t)_j - \widehat{\nabla}\mathcal{L}_{i_t}(\theta'_t)_j\right)\right)\right]
$$
$$
\leq \mathbb{E}\left[\left(\sqrt{2}\left\|\widehat{\nabla}\mathcal{L}_{i_t}(\theta_t)\right\| + \sqrt{2}\left\|\widehat{\nabla}\mathcal{L}_{i_t}(\theta'_t)\right\|\right)\left(\left\|\widehat{\nabla}\mathcal{L}_{i_t}(\theta_t) - \widehat{\nabla}\mathcal{L}_{i_t}(\theta'_t)\right\|\right)\right]
$$
$$
\leq 2\sqrt{2}dG\mathbb{E}\left[\left\|\widehat{\nabla}\mathcal{L}_{i_t}(\theta_t) - \widehat{\nabla}\mathcal{L}_{i_t}(\theta'_t)\right\|\right]
$$
$$
\leq 2\sqrt{2}dG\left(c_\mu L + \sqrt{3}L\Delta_t\right)
$$

The last two inequalities come from Lemma E.4 and E.1. Hence, we have

$$\mathbb{E}\left[\|v_{t+1} - v'_{t+1}\| | z_{i_t} = z_{i'_t}\right] \leq \beta_2 \Sigma_t + (1 - \beta_2) 2\sqrt{2} dG \left(c_\mu L + \sqrt{3} L \Delta_t\right)$$

Therefore, the expectation of $J_t$ under $z_{i_t} = z'_{i_t}$ is

$$\mathbb{E}[J_t] \leq dG \mathbb{E}\left[\left\|\frac{1}{\sqrt{v_{t+1} + \delta}} - \frac{1}{\sqrt{v'_{t+1} + \delta}}\right\|\right]$$

$$\leq \frac{dG}{2\delta^{3/2}} \mathbb{E}\left[\|v_{t+1} - v'_{t+1}\|\right]$$

$$\leq \frac{dG}{2\delta^{3/2}} \left(\beta_2 \Sigma_t + (1 - \beta_2) 2\sqrt{2} dG \left(c_\mu L + \sqrt{3} L \Delta_t\right)\right)$$

If we consider $\delta = \delta_0 n^{2/3} d^{2/3}$ and $1 - \beta_2 = \frac{1}{dt}$, we have

$$\mathbb{E}[J_t] \leq \frac{G}{2n\delta_0^{3/2}} \left(\beta_2 \Sigma_t + \frac{2\sqrt{2}G}{t} \left(c_\mu L + \sqrt{3} L \Delta_t\right)\right)$$

**Case 2:** With probability $\frac{1}{n}$, we have $z_{i_t} \neq z'_{i_t}$. Therefore, we obtain

$$\mathbb{E}[R_t] \leq \mathbb{E}\left[\left\|\frac{m_{t+1}}{\sqrt{v_{t+1} + \delta}}\right\|\right] + \mathbb{E}\left[\left\|\frac{m'_{t+1}}{\sqrt{v'_{t+1} + \delta}}\right\|\right]$$

$$\leq \frac{1}{\delta} \left(\mathbb{E}\left[\|m_{t+1}\|\right] + \mathbb{E}\left[\|m'_{t+1}\|\right]\right)$$

$$\leq \frac{2}{\delta} \left(\frac{c_\mu L}{2} + \sqrt{3}G\right)$$

$$= \frac{1}{\delta} \left(c_\mu L + 2\sqrt{3}G\right)$$

by Lemma E.2. Therefore, we have

$$\Delta_{t+1} \leq \Delta_t + \alpha \mathbb{E}[R_t | \mathcal{E}_{t_0}]$$

$$\leq \Delta_t + \alpha \left(1 - \frac{1}{n}\right) \frac{\beta_1 \Phi_t + (1 - \beta_1)\left(c_\mu L + \sqrt{3} L \Delta_t\right)}{\delta}$$

$$+ \alpha \left(1 - \frac{1}{n}\right) \frac{dG}{2\delta^{3/2}} \left(\beta_2 \Sigma_t + (1 - \beta_2) 2\sqrt{2} dG \left(c_\mu L + \sqrt{3} L \Delta_t\right)\right)$$

$$+ \frac{\alpha}{n\delta} \left(c_\mu L + 2\sqrt{3}G\right)$$

Arranging all the terms with respect to $\Delta_t$, $\Phi_t$, and $\Sigma_t$, we can rewrite the above inequality with the form as $\Delta_{t+1} \leq \mathcal{A}_t \Delta_t + \mathcal{B}_t \Phi_t + \mathcal{C}_t \Sigma_t + \mathcal{P}_t$ as

$$\Delta_{t+1} \leq \underbrace{\left[1 + \frac{\sqrt{3}\alpha(1 - \beta_1)L}{\delta} \left(1 - \frac{1}{n}\right) + \frac{\sqrt{6}d^2 G^2 \alpha(1 - \beta_2)L}{\delta^{3/2}} \left(1 - \frac{1}{n}\right)\right]}_{\mathcal{A}_t} \Delta_t$$

$$+ \underbrace{\frac{\alpha\beta_1}{\delta} \left(1 - \frac{1}{n}\right)}_{\mathcal{B}_t} \Phi_t$$

$$+ \underbrace{\frac{\alpha\beta_2 dG}{2\delta^{3/2}} \left(1 - \frac{1}{n}\right)}_{\mathcal{C}_t} \Sigma_t$$

$$+ \mathcal{P}_t$$

where $\mathcal{P}_t$ is defined by

$$\mathcal{P}_t = \frac{\alpha(1 - \beta_1)c_\mu L}{\delta} \left(1 - \frac{1}{n}\right) + \frac{\sqrt{2}\alpha(1 - \beta_2)c_\mu d^2 G^2 L}{\delta^{3/2}} \left(1 - \frac{1}{n}\right) + \frac{\alpha}{n\delta} \left(c_\mu L + 2\sqrt{3}G\right)$$

Under the parameter settings $\alpha_t = \frac{\alpha}{t}$, $1 - \beta_{1,t} = \frac{c_1}{nt}$, $1 - \beta_{2,t} = \frac{c_2}{dnt}$, and $\delta = \delta_0 d^{2/3}$, the quantity $\mathcal{P}_t$ is

$$\mathcal{P}_t = \frac{\alpha c_1 c_\mu L}{n \delta_0 t} \left( 1 - \frac{1}{n} \right) + \frac{\sqrt{2} \alpha c_2 c_\mu G^2 L}{n \delta_0^{3/2} t} \left( 1 - \frac{1}{n} \right) + \frac{\alpha}{n d^{2/3} \delta_0} \left( c_\mu L + 2\sqrt{3} G \right)$$

$$= \frac{1}{nt} \left( \frac{\alpha c_1 c_\mu L}{d^{2/3} \delta_0} \left( 1 - \frac{1}{n} \right) + \frac{\sqrt{2} \alpha c_2 c_\mu G^2 L}{\delta_0^{3/2}} \left( 1 - \frac{1}{n} \right) + \frac{\alpha}{d^{2/3} \delta_0} \left( c_\mu L + 2\sqrt{3} G \right) \right)$$

$\square$

**Lemma E.7** (Recursive Relation for $\Phi_t$)**.** *For any $t > t_0$, the quantity $\Phi_t$ has the following recursive relation*

$$\Phi_{t+1} \leq \beta_1 \Phi_t + (1 - \beta_1) \left( 1 - \frac{1}{n} \right) \left( c_\mu L + \sqrt{3} L \Delta_t \right) + \frac{1 - \beta_1}{n} \left( c_\mu L + 2\sqrt{3} G \right)$$

*Proof.* For any $t > t_0$, we have that

$$\|m_{t+1} - m'_{t+1}\| = \left\| \beta_1 m_t + (1 - \beta_1) \widehat{\nabla} \mathcal{L}(\theta_t; z_{i_t}) - \beta_1 m'_t - (1 - \beta_1) \widehat{\nabla} \mathcal{L}(\theta'_t; z'_{i_t}) \right\|$$

$$\leq \beta_1 \|m_t - m'_t\| + (1 - \beta_1) \left\| \widehat{\nabla} \mathcal{L}(\theta_t; z_{i_t}) - \widehat{\nabla} \mathcal{L}(\theta'_t; z'_{i_t}) \right\|$$

We again consider two cases.

**Case 1:** When $z_{i_t} = z'_{i_t}$ with probability $1 - \frac{1}{n}$. By Lemma E.1, it could hold that

$$\mathbb{E}_u \left[ \left\| \widehat{\nabla} \mathcal{L}(\theta_t; z_{i_t}) - \widehat{\nabla} \mathcal{L}(\theta'_t; z_{i_t}) \right\| \right] \leq \mu L (d+3)^{3/2} + \sqrt{3} L \|\theta_t - \theta'_t\|$$

Therefore, we have

$$\mathbb{E} \left[ \left\| \widehat{\nabla} \mathcal{L}(\theta_t; z_{i_t}) - \widehat{\nabla} \mathcal{L}(\theta'_t; z_{i_t}) \right\| \right] \leq c_\mu L + \sqrt{3} L \Delta_t$$

**Case 2:** When $z_{i_t} \neq z'_{i_t}$ with probability $\frac{1}{n}$. By Lemma E.2, it should hold that

$$\mathbb{E}_u \left[ \left\| \widehat{\nabla} \mathcal{L}(\theta_t; z_{i_t}) - \widehat{\nabla} \mathcal{L}(\theta'_t; z'_{i_t}) \right\| \right] \leq \mathbb{E} \left[ \left\| \widehat{\nabla} \mathcal{L}(\theta_t; z_{i_t}) \right\| \right] + \mathbb{E} \left[ \left\| \widehat{\nabla} \mathcal{L}(\theta'_t; z'_{i_t}) \right\| \right]$$

$$\leq 2 \left( \frac{c_\mu L}{2} + \sqrt{3} G \right)$$

$$= c_\mu L + 2\sqrt{3} G$$

by $G$-Lipschitz condition of $\mathcal{L}_j$ for $j \in [n]$. Thus, we have

$$\Phi_{t+1} \leq \beta_1 \Phi_t + (1 - \beta_1) \left( 1 - \frac{1}{n} \right) \left( c_\mu L + \sqrt{3} L \Delta_t \right) + \frac{1 - \beta_1}{n} \left( c_\mu L + 2\sqrt{3} G \right)$$

$$= \underbrace{(1 - \beta_1) \left( 1 - \frac{1}{n} \right) \sqrt{3} L}_{\mathcal{D}_t} \Delta_t$$

$$+ \underbrace{\beta_1}_{\mathcal{E}_t} \Phi_t$$

$$+ \underbrace{0}_{\mathcal{F}_t} \cdot \Sigma_t$$

$$+ \mathcal{Q}_t$$

where $\mathcal{Q}_t$ is computed as

$$\mathcal{Q}_t = (1 - \beta_1) \left( 1 - \frac{1}{n} \right) c_\mu L + \frac{1 - \beta_1}{n} \left( c_\mu L + 2\sqrt{3} G \right)$$

$$= (1 - \beta_1) c_\mu L + \frac{2\sqrt{3} (1 - \beta_1) G}{n}$$

Under the parameter settings $1 - \beta_{1,t} = \frac{c_1}{nt}$, the quantity $\mathcal{Q}_t$ is

$$\mathcal{Q}_t = \frac{c_1 c_\mu L}{nt} + \frac{2\sqrt{3} c_1 G}{n^2 t}$$

$$= \frac{1}{nt} \left( c_1 c_\mu L + \frac{2\sqrt{3} c_1 G}{n} \right)$$

$\square$

Finally, we compute the bound for $\Sigma_{t+1}$.

**Lemma E.8** (Recursive Relation for $\Sigma_t$). *For any $t > t_0$, the quantity $\Sigma_t$ has the following recursive relation*

$$\Sigma_{t+1} \le \beta_2 \Sigma_t + 9\left(1 - \frac{1}{n}\right)(1-\beta_2)C_1\left(\sqrt{2}\mu L(d+6)^{3/2} + \sqrt{6}L\Delta_t\right)$$

$$+ (1-\beta_2)\left(\frac{3\mu^2 L^2 (d+6)^3}{4} + 18G^2\right)$$

*Proof.* By the definition, we have

$$\|v_{t+1} - v'_{t+1}\| \le \beta_2 \|v_t - v'_t\| + (1-\beta_2)\left\|\widehat{\nabla}\mathcal{L}_{i_t}(\theta_t)^2 - \widehat{\nabla}\mathcal{L}_{i'_t}(\theta'_t)^2\right\|$$

We consider two cases.

**Case 1:** When $z_{i_t} = z'_{i_t}$ with probability $1 - \frac{1}{n}$. In this case, we have

$$\mathbb{E}\left[\|v_{t+1} - v'_{t+1}\|\right] \le \beta_2 \Sigma_t + (1-\beta_2)2\sqrt{2}dG\left(c_\mu L + \sqrt{3}L\Delta_t\right)$$

**Case 2:** When $z_{i_t} \neq z'_{i_t}$ with probability $\frac{1}{n}$. We have

$$\left\|\widehat{\nabla}\mathcal{L}_{i_t}(\theta_t)^2 - \widehat{\nabla}\mathcal{L}_{i'_t}(\theta'_t)^2\right\| \le \left\|\widehat{\nabla}\mathcal{L}_{i_t}(\theta_t)^2\right\| + \left\|\widehat{\nabla}\mathcal{L}_{i'_t}(\theta'_t)^2\right\|$$

$$\le \left[\sum_{j=1}^d \widehat{\nabla}\mathcal{L}_{i_t}(\theta_t)_j^4\right]^{1/2} + \left[\sum_{j=1}^d \widehat{\nabla}\mathcal{L}_{i'_t}(\theta'_t)_j^4\right]^{1/2}$$

$$\le \left[\sum_{j=1}^d \widehat{\nabla}\mathcal{L}_{i_t}(\theta_t)_j^2\right] + \left[\sum_{j=1}^d \widehat{\nabla}\mathcal{L}_{i'_t}(\theta'_t)_j^2\right]$$

$$= \left\|\widehat{\nabla}\mathcal{L}_{i_t}(\theta_t)\right\|^2 + \left\|\widehat{\nabla}\mathcal{L}_{i'_t}(\theta'_t)\right\|^2$$

By Lemma E.2, the expectation of above inequality would be

$$\mathbb{E}\left[\left\|\widehat{\nabla}\mathcal{L}_{i_t}(\theta_t)^2 - \widehat{\nabla}\mathcal{L}_{i'_t}(\theta'_t)^2\right\|\right] \le 2\left(\frac{3}{8}c_\mu^2 L^2 + 9G^2\right)$$

$$\le \frac{3}{4}c_\mu^2 L^2 + 18G^2$$

Finally, we have

$$\Sigma_{t+1} \le \beta_2 \Sigma_t + \left(1 - \frac{1}{n}\right)(1-\beta_2)2\sqrt{2}dG\left(c_\mu L + \sqrt{3}L\Delta_t\right) + \frac{1}{n}(1-\beta_2)\left(\frac{3}{4}c_\mu^2 L^2 + 18G^2\right)$$

$$= \underbrace{2\sqrt{6}(1-\beta_2)dGL\left(1 - \frac{1}{n}\right)}_{\mathcal{G}_t}\Delta_t$$

$$+ \underbrace{0}_{\mathcal{H}_t} \cdot \Phi_t$$

$$+ \underbrace{\beta_2}_{\mathcal{I}_t}\Sigma_t$$

$$+ \mathcal{R}_t$$

where $\mathcal{R}_t$ is defined by

$$\mathcal{R}_t = 2\sqrt{2}\left(1 - \frac{1}{n}\right)(1-\beta_2)c_\mu dGL + \frac{1-\beta_2}{n}\left(\frac{3}{4}c_\mu^2 L^2 + 18G^2\right)$$

Under the parameter settings $\alpha_t = \frac{\alpha}{t}$, $1 - \beta_{1,t} = \frac{c_1}{nt}$, $1 - \beta_{2,t} = \frac{c_2}{dt}$, and $\delta = \delta_0\sqrt{d}$, the quantity $\mathcal{R}_t$ becomes

$$\mathcal{R}_t = \frac{1}{nt}\left(2\sqrt{2}c_\mu GL\left(1 - \frac{1}{n}\right) + \frac{3c_\mu^2 L^2}{4dn} + \frac{18G^2}{dn}\right)$$

$\square$

### E.3 SPECTRAL NORM OF MATRIX $\Lambda_t$

We construct the key matrix $\Lambda_t$ for our recursive relations for $(\Delta_t, \Phi_t, \Sigma_t)$ as follows.

$$
\begin{bmatrix} \Delta_{t+1} \\ \Phi_{t+1} \\ \Sigma_{t+1} \end{bmatrix} = \underbrace{\begin{bmatrix} \mathcal{A}_t & \mathcal{B}_t & \mathcal{C}_t \\ \mathcal{D}_t & \mathcal{E}_t & \mathcal{F}_t \\ \mathcal{G}_t & \mathcal{H}_t & \mathcal{I}_t \end{bmatrix}}_{\Lambda_t} \begin{bmatrix} \Delta_t \\ \Phi_t \\ \Sigma_t \end{bmatrix} + \underbrace{\begin{bmatrix} \mathcal{P}_t \\ \mathcal{Q}_t \\ \mathcal{R}_t \end{bmatrix}}_{\Gamma_t}
$$

where each entry of $\Lambda_t$ is defined by

$$
\mathcal{A}_t = \left[ 1 + \frac{\sqrt{3}\alpha_t(1 - \beta_{1,t})L}{\delta}\left(1 - \frac{1}{n}\right) + \frac{\sqrt{6}d^2G^2\alpha_t(1-\beta_{2,t})L}{\delta^{3/2}}\left(1 - \frac{1}{n}\right) \right]
$$

$$
\leq \left[ 1 + \frac{\sqrt{3}\alpha c_1 L}{\delta_0 d^{2/3}nt^2} + \frac{\sqrt{6}\alpha c_2 G^2 L}{\delta_0^{3/2}t^2} \right]
$$

$$
\mathcal{B}_t = \frac{\alpha_t\beta_{1,t}}{\delta}\left(1 - \frac{1}{n}\right) \leq \frac{\alpha_t\beta_{1,t}}{\delta} = \frac{\alpha\beta_{1,t}}{\delta_0 d^{2/3}t}
$$

$$
\mathcal{C}_t = \frac{\alpha_t\beta_{2,t}dG}{2\delta^{3/2}}\left(1 - \frac{1}{n}\right) \leq \frac{\alpha_t\beta_{2,t}dG}{2\delta^{3/2}} = \frac{\alpha\beta_{2,t}G}{2\delta_0^{3/2}t}
$$

$$
\mathcal{D}_t = (1 - \beta_{1,t})\left(1 - \frac{1}{n}\right)\sqrt{3}L \leq \frac{\sqrt{3}c_1 L}{nt}
$$

$$
\mathcal{E}_t = \beta_{1,t} \leq 1
$$

$$
\mathcal{F}_t = 0
$$

$$
\mathcal{G}_t = 2\sqrt{6}(1 - \beta_{2,t})dGL\left(1 - \frac{1}{n}\right) \leq \frac{2\sqrt{6}c_2 GL}{nt}
$$

$$
\mathcal{H}_t = 0
$$

$$
\mathcal{I}_t = \beta_{2,t} \leq 1
$$

under our parameter configurations. The most complicated entries are $\mathcal{A}_t, \mathcal{B}_t$, and $\mathcal{C}_t$. To guarantee the uniform stability of ZO-Adam, we let

1. $\alpha_t \leftarrow \dfrac{\alpha}{t}$ (diminishing learning rate)

2. $\beta_{1,t} = 1 - \dfrac{\beta_1}{nt}$

3. $\beta_{2,t} = 1 - \dfrac{\beta_2}{dnt}$

**Lemma E.9.** *It should hold that*

$$
(\mathcal{A}_t v_1 + \mathcal{B}_t v_2 + \mathcal{C}_t v_3)^2 \leq v_1^2 + \frac{1}{t}U_1 + \frac{1}{t^2}U_2
$$

$$
(\mathcal{D}_t v_1 + \mathcal{E}_t v_2 + \mathcal{F}_t v_3)^2 \leq v_2^2 + \frac{1}{t}V_1
$$

$$
(\mathcal{G}_t v_1 + \mathcal{H}_t v_2 + \mathcal{I}_t v_3)^2 \leq v_3^2 + \frac{1}{t}T_1
$$

*for some suitable constants $U_1, U_2, V_1$, and $T_1$.*

*Proof.* By the definition of each constant, we have

$$
\mathcal{A}_t^2 = 1 + \frac{\alpha}{t^2}A_1 + \frac{\alpha^2}{t^4}A_2
$$

$$
\mathcal{B}_t^2 \leq \frac{\alpha^2}{t^2\delta_0^2}
$$

$$
\mathcal{C}_t^2 \leq \frac{\alpha^2(L + 4dG)^2}{8t^2\delta^4\gamma}\left(1 - \frac{1}{n}\right)^2
$$

Hence, we can write the following term as

$$(\mathcal{A}_t v_1 + \mathcal{B}_t v_2 + \mathcal{C}_t v_3)^2 = \mathcal{A}_t^2 v_1^2 + \underbrace{(\mathcal{B}_t v_2 + \mathcal{C}_t v_3)^2}_{\text{Depends on } \frac{\alpha^2}{t^2}} + \underbrace{2\mathcal{A}_t v_1(\mathcal{B}_t v_2 + \mathcal{C}_t v_3)}_{\text{Depends on } \frac{\alpha}{t} \text{ and } \frac{\alpha^2}{t^2}}$$

$$\leq v_1^2 + \frac{1}{t}U_1 + \frac{1}{t^2}U_2 + \frac{1}{t^3}U_3 + \frac{1}{t^4}U_4$$

since $v_i^2 \leq 1$ for each $i \in \{1, 2, 3\}$. The second row would be

$$(\mathcal{D}_t v_1 + \mathcal{E}_t v_2 + \mathcal{F}_t v_3)^2 = \left((1 - \beta_1)\left(1 - \frac{1}{n}\right)\sqrt{3}Lv_1 + \beta_1 v_2\right)^2$$

$$\leq (1 - \beta_{1,t})3L^2\left(1 - \frac{1}{n}\right)^2 v_1^2 + \beta_{1,t}v_2^2$$

$$\leq v_2^2 + \frac{1}{t}V_1$$

where we use Jensen's inequality for convex function $f(x) = x^2$ and $1 - \beta_{1,t} = \frac{\beta_1}{t}$. The last row can be computed as

$$(\mathcal{G}_t v_1 + \mathcal{H}_t v_2 + \mathcal{I}_t v_3)^2 \leq \left(2\sqrt{6}(1 - \beta_{2,t})dGL\left(1 - \frac{1}{n}\right)v_1 + \beta_2 v_3\right)^2$$

$$\leq \left(\frac{2\sqrt{6}GL}{nt}v_1 + \beta_2 v_3\right)^2$$

$$\leq v_3^2 + \frac{1}{t}T_1 + \frac{1}{t^2}T_2$$

where $T_1$ is dimension-free since $1 - \beta_{2,t} = \frac{1}{dnt}$. $\qquad\square$

**Lemma E.10.** *The spectral norm of the matrix $\Lambda_t$ is upper-bounded by*

$$\left\|\!\left\|\Lambda_t\right\|\!\right\|_2 \leq \exp\left(\frac{1}{t}W_1 + \frac{1}{t^2}W_2 + \frac{1}{t^3}W_3 + \frac{1}{t^4}W_4\right)$$

*Proof.* By above Lemma, we have for suitable constants $\{W_i\}_{i=1}^4$

$$\|\Lambda_t v\|^2 \leq (v_1^2 + v_2^2 + v_3^2) + \frac{1}{t}W_1 + \frac{1}{t^2}W_2$$

$$\leq 1 + \frac{1}{t}W_1 + \frac{1}{t^2}W_2 + \frac{1}{t}W_3 + \frac{1}{t^4}W_4$$

$$\leq \exp\left(\frac{1}{t}W_1 + \frac{1}{t^2}W_2 + \frac{1}{t^3}W_3 + \frac{1}{t^4}W_4\right)$$

where we use the inequality $1 + x \leq e^x$ for $x \geq 0$. $\qquad\square$

**Lemma E.11.** *The norm of vector $\Gamma_t$ is bounded by*

$$\|\Gamma_t\|^2$$

$$\leq \frac{3}{n^2 t^2}\left(\frac{\alpha^2 c_1^2 c_\mu^2 L^2}{d^{4/3}\delta_0^2} + \frac{2\alpha c_2^2 c_\mu^2 G^4 L^2}{\delta_0^3} + \frac{\alpha^2\left(c_\mu L + 2\sqrt{3}G\right)^2}{d^{4/3}\delta_0^2} + c_1^2 c_\mu^2 L^2 + \frac{12 c_1^2 G^2}{n^2} + 8c_\mu^2 G^2 L^2 + \frac{9c_\mu^4 L^4}{16d^2 n^2} + \frac{324 G^4}{d^2 n^2}\right)$$

*Proof.* By the definition of $\Gamma_t$, we have

$$\|\Gamma_t\|^2 = \mathcal{P}_t^2 + \mathcal{Q}_t^2 + \mathcal{R}_t^2$$

$$\leq \frac{1}{n^2 t^2}\left(\frac{\alpha c_1 c_\mu L}{d^{2/3}\delta_0}\left(1 - \frac{1}{n}\right) + \frac{\sqrt{2}\alpha c_2 c_\mu G^2 L}{\delta_0^{3/2}}\left(1 - \frac{1}{n}\right) + \frac{\alpha}{d^{2/3}\delta_0}\left(c_\mu L + 2\sqrt{3}G\right)\right)^2$$

$$+ \frac{1}{n^2 t^2}\left(c_1 c_\mu L + \frac{2\sqrt{3}c_1 G}{n}\right)^2$$

$$+ \frac{1}{n^2 t^2}\left(2\sqrt{2}c_\mu GL\left(1 - \frac{1}{n}\right) + \frac{3c_\mu^2 L^2}{4dn} + \frac{18G^2}{dn}\right)^2$$

Rearranging all the terms, we have

$$\|\Gamma_t\|^2$$

$$\leq \frac{3}{n^2 t^2}\left(\frac{\alpha^2 c_1^2 c_\mu^2 L^2}{d^{4/3}\delta_0^2} + \frac{2\alpha c_2^2 c_\mu^2 G^4 L^2}{\delta_0^3} + \frac{\alpha^2 \left(c_\mu L + 2\sqrt{3}G\right)^2}{d^{4/3}\delta_0^2} + c_1^2 c_\mu^2 L^2 + \frac{12 c_1^2 G^2}{n^2} + 8 c_\mu^2 G^2 L^2 + \frac{9 c_\mu^4 L^4}{16 d^2 n^2} + \frac{324 G^4}{d^2 n^2}\right)$$

where $Z$ is well-defined by the above constant. $\qquad\square$

**Lemma E.12.** *We let*

$$M_t = \left\| \begin{bmatrix} \Delta_t \\ \Phi_t \\ \Sigma_t \end{bmatrix} \right\|$$

*Then, we have*

$$M_{T+1} \leq \frac{Z \exp\left(\zeta_2 W_2 + \zeta_3 W_3 + \zeta_4 W_4\right)}{n}\frac{1}{\alpha W_1}\left(\frac{T}{t_0}\right)^{\alpha W_1}$$

*Proof.* We let

$$\zeta_s = \sum_{n=1}^{\infty}\frac{1}{n^s}$$

It is well-known that $\zeta_s$ is finite for the integer $s \geq 2$. By the recursive relation in Lemma 1 and 2, we have

$$M_{T+1} \leq \sum_{t=t_0+1}^{T}\left[\prod_{k=t+1}^{T}\exp\left(\frac{\alpha}{k}W_1 + \frac{\alpha^2}{k^2}W_2\right)\right] \times \frac{Z}{nt}$$

$$= \frac{1}{n}Z\sum_{t=t_0+1}^{T}\left[\prod_{k=t+1}^{T}\exp\left(\frac{\alpha}{k}W_1 + \frac{\alpha^2}{k^2}W_2\right)\right]\frac{1}{t}$$

$$\leq \frac{1}{n}Z\sum_{t=t_0+1}^{T}\left[\exp\left(\alpha W_1\sum_{k=t+1}^{T}\frac{1}{k} + \alpha^2 W_2\sum_{k=t+1}^{T}\frac{1}{k^2}\right)\right]\frac{1}{t}$$

$$\leq \frac{1}{n}Z\sum_{t=t_0+1}^{T}\left[\exp\left(\alpha W_1\log\left(\frac{T}{t}\right) + \alpha^2\zeta_2 W_2\right)\right]\frac{1}{t}$$

$$\leq \frac{Z}{n}\exp\left(\zeta_2 W_2 + \zeta_3 W_3 + \zeta_4 W_4\right)\sum_{t=t_0+1}^{T}\left(\frac{T}{t}\right)^{\alpha W_1}\frac{1}{t}$$

$$= \frac{Z\exp\left(\zeta_2 W_2 + \zeta_3 W_3 + \zeta_4 W_4\right)}{n}T^{\alpha W_1}\sum_{t=t_0+1}^{T}t^{-\alpha W_1-1}$$

$$\leq \frac{Z\exp\left(\zeta_2 W_2 + \zeta_3 W_3 + \zeta_4 W_4\right)}{n}T^{\alpha W_1}\int_{t_0}^{T}t^{-\alpha W_1-1}dt$$

$$\leq \frac{Z\exp\left(\zeta_2 W_2 + \zeta_3 W_3 + \zeta_4 W_4\right)}{n}\frac{1}{\alpha W_1}\left(\frac{T}{t_0}\right)^{\alpha W_1}$$

$\qquad\square$

Finally, we will bound the term $\Delta_{T+1}$.

*Proof.* The generalization error bound for ZO-Adam is

$$R(f(\cdot, z)) = \mathbb{E}\left[|f(\theta_T; z) - f(\theta'_T; z)|\right]$$

$$\leq 2M\frac{t_0}{n} + G\Delta_T$$

$$\leq \frac{2Mt_0}{n} + \frac{Z\exp\left(\zeta_2 W_2 + \zeta_3 W_3 + \zeta_4 W_4\right)}{n}\frac{G}{\alpha W_1}\left(\frac{T}{t_0}\right)^{\alpha W_1}$$

Therefore, we have

$$R(f(\cdot, z)) \leq \frac{1}{n} \left[ 2Mt_0 + \frac{ZG \exp\left(\zeta_2 W_2 + \zeta_3 W_3 + \zeta_4 W_4\right)}{\alpha W_1} \left(\frac{T}{t_0}\right)^{\alpha W_1} \right]$$

We consider the function

$$g(x) := C_1 x + C_2 \left(\frac{T}{x}\right)^{C_3}$$

To find the minimizer of this function, we take the derivative as

$$g'(x) = C_1 - C_2 C_3 \frac{T^{C_3}}{x^{C_3+1}}$$

Therefore, the function $g$ has a local optimum at

$$\widehat{x} = \left(\frac{C_2 C_3}{C_1}\right)^{\frac{1}{C_3+1}} T^{\frac{C_3}{C_3+1}}$$

The second derivative of this function would be

$$g''(x) = C_2 C_3 (C_3 + 1) \frac{T^{C_3}}{x^{C_3+2}} > 0$$

for $x > 0$. Therefore, the local optimum $\widehat{x}$ is the minimizer of the function $g$. Therefore, the function $g$ has a minimal value as

$$g(\widehat{x}) = C_1 \left(\frac{C_2 C_3}{C_1}\right)^{\frac{1}{C_3+1}} T^{\frac{C_3}{C_3+1}} + C_2 \left(T \left(\frac{C_1}{C_2 C_3}\right)^{\frac{1}{C_3+1}} T^{-\frac{C_3}{C_3+1}}\right)^{C_3}$$

$$= C_1 \left(\frac{C_2 C_3}{C_1}\right)^{\frac{1}{C_3+1}} T^{\frac{C_3}{C_3+1}} + C_2 \left(\frac{C_1}{C_2 C_3}\right)^{\frac{C_3}{C_3+1}} T^{\frac{C_3}{C_3+1}}$$

$$= \left[ C_1 \left(\frac{C_2 C_3}{C_1}\right)^{\frac{1}{C_3+1}} + C_2 \left(\frac{C_1}{C_2 C_3}\right)^{\frac{C_3}{C_3+1}} \right] T^{\frac{C_3}{C_3+1}}$$

Under $C_1 = 2M$, $C_2 = \dfrac{ZG \exp\left(\zeta_2 W_2 + \zeta_3 W_3 + \zeta_4 W_4\right)}{\alpha W_1}$, and $C_3 = \alpha W_1$, we have

$$\epsilon_{\text{gen}} \leq \mathcal{O}\left(\frac{T^r}{n}\right)$$

for some positive constant $r < 1$. Note that $r \propto \frac{d}{\delta^{3/2}}$ since the constants $W_i \propto \frac{d}{\delta^{3/2}}$. $\qquad\square$

