# OpenReview forum: "MeZO-A$^{3}$dam: Memory-efficient Zeroth-order Adam with Adaptivity Adjustments for Fine-tuning LLMs"
_ICLR.cc/2025/Conference — ICLR 2025 Conference Withdrawn Submission_

### Official Review · Reviewer_3fTk · 2024-10-31

**Soundness:** 2
**Presentation:** 2
**Contribution:** 2
**Rating:** 3
**Confidence:** 4

**Summary:**

The paper claims that $\delta$ in MeZO-Adam is very important for the zero-order optimization method. Through theoretical analysis, an approach of setting $\delta$ is given, which is related to the dimension of model parameters. Experiments test the effectiveness of the proposed approach.

**Strengths:**

- The idea is simple and clear.  The paper finds that $\delta$ in MeZO-Adam is very important for the zero-order optimization method. This gives an idea of dealing with the large variance of the estimated gradients in MeZO-Adam.

- This paper gives an approach of setting $\delta$ from the perspective of generalization theory.

- Experiments on LLM fine-tuning tasks verify the superiority of the proposed approach. Compared with MeZO, the performance is significantly improved with a small amount of memory. In addition, the source code of the paper is submitted in the attachment, which is helpful to reproduce the results of the paper.

**Weaknesses:**

- I question the proposed method's dimension-free claim, as the theorem appears problematic. Specifically, the proof of Lemma C.6 is incorrect, although I haven't verified all the theorem's proofs. There is a missing of the model dimension d. It is unknown whether the theorem can be proven to exclude d if this lemma includes d. See the same proof of Lemma 7 in the following paper:

Subzero: Random Subspace Zeroth-Order Optimization for Memory-Efficient LLM Fine-Tuning

https://arxiv.org/pdf/2410.08989

- The paper should provide more practical verification of the proposed method, such as a setting for $\delta$ applicable to any model, rather than focusing heavily on theoretical analysis. The paper needs a method for setting $\delta$ across all models without requiring adjustments when model dimensions change. Currently, it necessitates different $\delta$ setting for each network. Also, no ablation experiment on $\delta$ has been conducted. The performance variations with different $\delta$ settings are not addressed. Additionally, the $\delta$ setting is influenced by the model dimension d, with no ambiguity regarding full parameter fine-tuning. However, it does not clarify whether the $\delta$ setting method should depend on the parameter dimension d or the dimension of the learnable parameters in parameter-efficient fine-tuning schemes, e.g., LoRA.

- The experimental results are weak, as the networks are small and single-task. LLM fine-tuning generally focuses on large models, such as model parameters exceeding 7B. The GPUs used in this paper are all advanced ones, which are fully capable of fine-tuning models above 7B. Why did not the authors do it? Since the paper relies on MeZO, it is a reasonable choice to repeat its experiments.

**Questions:**

I have the following comments. It is hoped that the quality of the paper can be improved.

- Whether all the compared methods use the same batch size. If the batch sizes are different, the comparison is unfair, because the larger the batch size, the smaller the variance of the estimated gradients. Since the paper directly cites experimental results from other papers, it is essential to confirm whether all compared methods utilized the same batch size. Otherwise, the experiments need to be done again.

- It needs to discuss the differences with the paper of (Chen et al., 2019), including why their approach cannot be applied to prove the convergence of MeZO-$\text{A}^3$dam, as well as the challenges involved.

- Some empirical work [1,2] indicates that ZO-Adam is not better than ZO-SGD. Since the paper proves the convergence of ZO-Adam, it should  compare with ZO-SGD thoroughly.

[1] Yihua Zhang, Pingzhi Li, Junyuan Hong, Jiaxiang Li, Yimeng Zhang, Wenqing Zheng, Pin-Yu Chen, Jason D. Lee, Wotao Yin, Mingyi Hong, Zhangyang Wang, Sijia Liu, and Tianlong Chen. Revisiting zeroth-order optimization for memory-efficient LLM fine-tuning: A benchmark. ICML, 2024.

[2] Wentao Guo, Jikai Long, Yimeng Zeng, Zirui Liu, Xinyu Yang, Yide Ran, Jacob R Gardner, Osbert Bastani, Christopher De Sa, Xiaodong Yu, Beidi Chen, Zhaozhuo Xu. Zeroth-order fine-tuning of LLMs with extreme sparsity. arXiv:2406.02913, 2024.

- It should test with other parameter efficient fine-tuning schemes: LoRA (Hu et al., 2022), prefix tuning (Li & Liang, 2021), and
prompt tuning (Lester et al., 2021).

- The paper introduces a method for setting delta in Adam based on model dimension d, defined as $\delta = \delta_0 * h(d) = \delta_0 * s * d^{2/3}$, where $s$ ranges from 0.1 to 10 and $\delta_0 = 10^{-8}$. How should the hyperparameter $s$ be set in experiments? Regarding $s$, it needs an ablation experiment to test the performance on different networks and datasets.

- It should give the results of the first-order methods SGD and Adam as references.

- The proof of Lemma C.6 is not correct. It should be (d+2) rather than 3 in the conclusion.

- Prompt-free is meaningful only when the model parameters are large, e.g., over 7B.  However, almost all models over 7B have been fine-tuned using prompt templates, which will definitely help the model to better understand the problem, and fine-tuning can converge faster. The paper is suggested to adopt the experimental setups of MeZO and select larger models.

---

### Official Review · Reviewer_e3uJ · 2024-11-01

**Soundness:** 2
**Presentation:** 3
**Contribution:** 3
**Rating:** 5
**Confidence:** 3

**Summary:**

The paper proposed a new ZO optimizer by introducing an adaptive adjustment parameter $\delta$ to the optimizer to improve the convergence and final performance. Through experiments on LLMs, the proposed method shows better performance and faster convergence in finetuning.

**Strengths:**

1. The proposed method shows better performance than the baselines.
2. The proposed method integrates theoretical support and the generalization analysis.

**Weaknesses:**

Though interesting, my major concern is if the same performance can be achieved by baselines by tuning $\alpha$, $\beta_1$ and $\beta_2$.

**Questions:**

1. Should vector $u$ on the denominator in Definition 2.1? If $i$th element $u(i)$ is really large by chance and the first term on the RHS multiply $u(i)$ again, this estimated gradient could explode.
2. "Note that, in Alg. 1, the construction of $m_t$ and $v_t$ could be implemented in a memory-efficient manner, however, we empirically observe that it is not computationally efficient." What does this refer to?
3. How is the proposed $\delta$ essentially different from using a smaller learning rate $\alpha$, $\beta_1$, $\beta_2$? They will also reduce the issue from the high variance of stochastic ZO gradient estimation.

---

### Official Review · Reviewer_H481 · 2024-11-02

**Soundness:** 3
**Presentation:** 4
**Contribution:** 3
**Rating:** 6
**Confidence:** 4

**Summary:**

This paper proposes MeZO-A3dam, based on the weak adaptivity hypothesis, which adjusts the adaptivity according to the parameter dimension. Theoretical and empirical evaluation validates its effectiveness over other zeroth-order baselines.

**Strengths:**

- This paper proposes a theoretical hypothesis and further an optimizer based on it, making it more solid and insightful.
- This paper is well-written, with clear motivation and illustrations.
- Figures and tables are clear and easy to read.

**Weaknesses:**

- Despite its superiority over other zeroth-order optimizer baselines, the widely used first-order baselines, e.g., SGD and Adam, are missing. This comparison is important for understanding the trade-offs between memory savings, performance loss, and real-world applications.
- The paper lacks comprehensive ablation studies on different components of their method. For example, they don't thoroughly explore different scaling functions h(d) for adaptivity adjustment, despite its importance to their method.

**Questions:**

See weaknesses.

---

### Official Review · Reviewer_Sd2v · 2024-11-03

**Soundness:** 3
**Presentation:** 3
**Contribution:** 3
**Rating:** 5
**Confidence:** 5

**Summary:**

The paper proposes MeZO-A3dam, a memory-efficient adaptive gradient ZO method for finetuning LLMs. Motivated by the limitations of existing ZO optimizers, such as slow convergence and reliance on handcrafted prompts, the authors investigate adaptive gradients and introduce the "weak adaptivity hypothesis." This hypothesis suggests that reducing adaptivity, scaled with parameter dimensions, improves optimization efficiency. MeZO-A3dam addresses high gradient variance in ZO settings by adjusting adaptivity according to model size, achieving dimension-free convergence and generalization guarantees. Experiments on various models and tasks demonstrate MeZO-A3dam's good convergence speed, generalization, and memory efficiency, significantly outperforming ZO baselines like MeZO-SVRG and MeZO-Adam​.

**Strengths:**

This paper has the following strengths:

* The proposed weak adaptivity hypothesis enables dimension-free convergence and generalization guarantees, which are rarely provided in zeroth-order optimization.

* MeZO-A3dam introduces a novel adaptivity scaling mechanism based on model parameter dimensions, addressing the high variance in ZO gradient estimates. This adjustment optimizes the convergence rate for large models.

* The paper is well-written and easy to follow.

**Weaknesses:**

This paper has the following weaknesses:

* I would suggest the authors make the color system in the plottings consistent, as the conflicting colors in Figure 2, Figure 3(a) and Figure 3(b) is confusing at first sight.

* The authors constantly mention the "hand-crafted prompts" is a weakness of prior arts (Line 17, Line 66, Line 721). I am wondering does the method proposed in this work solve this problem? Did the authors mention it somewhere in the paper? I did not find the relevant discussion so I got a bit confused.

* The 8-bit trick can be used to all the methods equipped with Adam, right? I would suggest the authors report the performance/efficiency of 8-bit MeZO-Adam.

* In the MeZO paper, the authors discussed a lot of PEFT adds-on for ZO method, which can further save a lot of memory while maintaining high-level performance as well. I am wondering if the authors have tried similar experiments in this work? I guess the advantage brought by the adaptivity will be greatly hurt if PEFT modules are applied, right? This is because the parameter space $d$ is much smaller, if LoRA, for example, is used.

**Questions:**

I do not have other questions. Please refer to the weaknesses of this paper.

---

### Note · Authors · 2024-11-26

**Comment:**

Dear AC and Reviewers,

Thank you for your valuable feedback and insightful comments on our submission. We deeply appreciate the time and effort you dedicated to reviewing our work.

After carefully considering the reviewers' comments and conducting additional experiments, we have identified areas of the work that require further refinement and deeper exploration. While we remain confident in the potential of our approach, we believe that addressing these points thoroughly will result in a stronger and more robust contribution to the field.

In light of this, we have decided to withdraw our submission at this time to allow us the opportunity to revisit and strengthen the work. This decision reflects our commitment to ensuring the highest quality in our research.

We are grateful for the constructive feedback provided during this process, which has been invaluable in shaping our future direction. We hope to re-engage with the community with an improved version of this research in the near future,

Thank you for your understanding and support.

**Withdrawal Confirmation:**

I have read and agree with the venue's withdrawal policy on behalf of myself and my co-authors.